



# Six years of greenhouse gas fluxes at Saclay, France, estimated with the Radon Tracer Method

Camille Yver-Kwok[1], Michel Ramonet[1], Leonard Rivier[1], Jinghui Lian[1,2], Claudia Grossi[3], Roger Curcoll[3], Dafina Kikaj[4], Edward Chung[4], and Ute Karstens[5]

[1]Laboratoire des Sciences du Climat et de l'Environnement (LSCE-IPSL), CEA-CNRS-UVSQ, Université Paris-Saclay, F-91191 Gif-sur-Yvette, France
[2]Origins.S.A.S, Suez Group, Tour CB21, 16 Place de l'Iris, 92040 Paris La Défense, Cedex, France
[3]Institut Català de Ciències del Clima (IC3), Barcelona, Spain
[4]National Physical Laboratory, Teddington, Middlesex, United Kingdom
[5]ICOS ERIC Carbon Portal, Physical Geography and Ecosystem Science,Lund University, Sölvegatan 12, 22362 Lund, Sweden

**Correspondence:** Camille Yver-Kwok (camille.yver@lsce.ipsl.fr)

**Abstract.** Here, we use carbon dioxide ($CO_2$), methane ($CH_4$), carbon monoxide (CO), nitrous oxide ($N_2O$) and radon ($^{222}Rn$) data from the Saclay ICOS tall tower in France to estimate $CO_2$, $CH_4$ and CO fluxes within the station footprint from January 2017 to December 2022 and $N_2O$ fluxes from February 2019 to December 2022 using the Radon Tracer Method (RTM).

We first performed a sensitivity study of this method applied to $CH_4$ and combined with different radon exhalation maps
including the improved European process-based radon flux maps developed within 19ENV01 traceRadon and back-trajectories in order to optimize it. Then, radon exhalation maps from the 19ENV01 traceRadon project, STILT trajectories from the ICOS Carbon Portal, best estimate of radon activity concentration and greenhouse gas data have been used to estimate the surface emissions. To our knowledge, this is the first study using the latest radon exhalation maps and standardized radon measurements to estimate $CO_2$, $CH_4$, CO and $N_2O$ surface emissions. We found that the average RTM estimates are $609 \pm 402$ mg m$^{-2}$h$^{-1}$,
$0.81 \pm 0.66$ mg m$^{-2}$h$^{-1}$,$1.04\pm1.80$ mg m$^{-2}$h$^{-1}$ and $0.063 \pm 0.079$ mg m$^{-2}$h$^{-1}$ for $CO_2$, $CH_4$, CO and $N_2O$ respectively. These fluxes are in good agreement with the literature.

$CH_4$, $N_2O$ and CO are also in fair agreement with the inventories, though with higher values. $CO_2$ fluxes are about five times higher than modeled anthropogenic and biogenic fluxes combined. The differences mainly occur during summer, and the $CO/CO_2$ ratio points toward a misrepresentation of the biogenic fluxes at this time by the WRF-VPRM version used here.

# 1 Introduction

Precise greenhouse gas monitoring began with $CO_2$ in the 1960s at background stations such as the South Pole and the Mauna Loa Observatory (Keeling, 1960; Brown and Keeling, 1965; Pales and Keeling, 1965). Since then, $CO_2$ as well as other greenhouses gases (GHG) such as $CH_4$ and $N_2O$ levels have risen significantly in the atmosphere. To monitor these changes, measurement networks (Prinn et al., 2018; Andrews et al., 2014; Fang et al., 2014; Ramonet et al., 2010) and coordinating pro-
grams (WMO, 2014) have been developed worldwide and help disentangle the different roles of the biospheric fluxes, oceanic





fluxes and anthropogenic emissions. Initially, background stations were set up to monitor long-term changes in hemispheric mean GHG amount fraction globally. However, in recent years, there has been a shift towards measuring regional and national amount fraction to verify and improve GHG emission assessments, known as "bottom-up" methods. Bottom-up methods rely on aggregated activity data, emission factors, and facility-level measurements, but often have significant uncertainties, espe-

cially for non-$CO_2$ GHGs, due to varying emission factors across sectors and biases from unaccounted sources. This smaller scale is thus especially relevant in the context of monitoring and verifying the international climate agreements (Bergamaschi et al., 2018).

Within the existing networks, the Integrated Carbon Observation System (ICOS) is a pan-European research infrastructure (Heiskanen et al. (2021); Yver-Kwok et al. (2021), https://www.icos-ri.eu, last access: 03 October 2024) which provides highly

compatible, harmonized and high-precision scientific data on the carbon cycle and greenhouse gas budget. It began with a preparatory phase from 2008 to 2013 and a demonstration period until 2015 when ICOS officially started as a legal entity. Three monitoring networks, atmospheric observations, flux measurements within and above ecosystems, and measurements of $CO_2$ partial pressure in seawater contribute to ICOS. To date, the atmospheric network consists of 39 stations located mostly in Europe (https://www.icos-atc.lsce.ipsl.fr/network, last access: 03 October 2024) and seven more stations will join the

network in the years to come. Depending on station class (class 1 or 2), different parameters are mandatory or recommended for measurement. $CO_2$ and $CH_4$ are mandatory at all stations, CO is mandatory only for class 1, and $N_2O$ and $^{222}Rn$ are recommended observations (ICOS RI, 2020).

Combining regional atmospheric GHG measurements with atmospheric transport models provides an opportunity for independent "top-down" verification of GHG bottom-up estimates, known as inverse modelling (Bergamaschi et al., 2018).

However, due to the significant uncertainties associated with atmospheric transport models, this method is not yet fully reliable for verification of GHG fluxes. An alternative independent method for verifying GHG fluxes is the observation-based Radon Tracer Method (RTM). This method is relatively simple and does not require sophisticated atmospheric transport modelling. RTM involves simultaneous measurements of radon ($^{222}Rn$) and GHG at co-located sites, along with the estimation of a radon source function. Radon is particularly useful because it is a naturally occurring radioactive gas with a well-defined source

(soil), a simple sink (half-life of 3.82 days), and chemical inertness. Due to these properties, radon can be used effectively as a tool for estimating and verifying GHG fluxes (Chambers et al., 2019; Kikaj et al., 2020; Zhang et al., 2021; Quérel et al., 2022). Indeed, like GHGs which usually have their sources close to the ground, $^{222}Rn$ accumulates during the night within the lower boundary layer. Thus, both $^{222}Rn$ and GHGs will accumulate together overnight and their correlation can be used to estimate the flux of GHG as long as we know the exhalation rate of $^{222}Rn$. The RTM has been used in many studies to

evaluate the fluxes between the atmosphere and ecosystems of trace gases such as $CO_2$, $CH_4$, $N_2O$, $H_2$ or COS (e.g.: Levin et al. (1999); Schmidt et al. (2001); Biraud et al. (2000); Messager (2007); Yver et al. (2009); Hammer and Levin (2009); Lopez et al. (2012); Vogel et al. (2012); Belviso et al. (2013); Grossi et al. (2018); Belviso et al. (2020); Levin et al. (2021); Tong et al. (2023)). Historically, the RTM has been applied in two main ways: to investigate regional-scale fluxes on an event basis (where an event may span hours or days) or to investigate local-scale fluxes on a nocturnal basis.

In Levin et al. (2021), the limits of the method were thoroughly studied. Their conclusions are summarized here:





- – The reliability of total nocturnal GHG emission estimates with the RTM critically depends on the accuracy and representativeness of the $^{222}$Rn exhalation rates estimated from the soils in the footprint of the site.

- – Using older $^{222}$Rn flux maps such as estimated by López-Coto et al. (2013) or Karstens et al. (2015) could lead to differences in the estimated GHG fluxes as large as a factor of 2 depending on which map is used.

– RTM-based GHG flux estimates also depend on the parameters chosen for the nighttime correlations of GHG and $^{222}$Rn, such as the nighttime period for regressions and the $R^2$ cut-off value for the goodness of the fit.

Notwithstanding these points, the RTM shows a good potential to be used in ICOS where there is already 15 stations measuring radon, with a precision and resolution comparable to the GHG. Moreover, thanks to Kikaj et al. (2024), the radon data can be harmonized and optimized to obtain the best radon estimates.

Finally, until recent years, the radon flux was considered homogeneous over time and space and the zone of influence of the flux was calculated using wind speed (Yver et al., 2009). This has been the high limits of this method as it is now known that the radon fluxes variate on space and time. However, advancements have been made through process-based radon flux maps from Szegvary et al. (2009); López-Coto et al. (2013); Karstens et al. (2015), and the development of footprint models such as STILT or FLEXPART (Brioude et al., 2013; Nehrkorn et al., 2010). These advancements have refined the RTM, improving

its accuracy and reliability in assessing GHG fluxes. Within the traceRadon project (Röttger et al., 2021), these process-based maps have been improved using among others, last generation moisture models with an increased time and space resolution (Karstens and Levin, 2024).

In this paper, we use $CO_2$, $CH_4$, CO and $^{222}$Rn data from the Saclay ICOS tall tower in France to estimate the $CO_2$, $CH_4$ and CO fluxes within the station footprint between January 2017 and December 2022 and $N_2O$ and $^{222}$Rn data from February

2019 to December 2022. In section 2, we describe the site, measurement techniques and method. In section 3, a sensitivity test of the RTM is performed and analyzed for two months in 2019. Finally, in section 4, RTM estimated fluxes are discussed and compared to bottom-up emissions from several inventories as well as to the literature.

## 2  Methods

### 2.1  ICOS Saclay tall tower description

Saclay (SAC) is located 20 km south-west of Paris, France, 48.722°N, 2.142°E, 160 m above sea level. It is an ICOS class 1 tall tower (Yver-Kwok et al., 2021). A 3-month intercomparison of radon monitors was previously carried out at this site in 2016 (Grossi et al., 2019). The station is located within a nuclear research center and 1 km north of the nearest village, Saint-Aubin, 680 inhabitants. A large university campus is located 2 km to the southeast of the site, with buildings still in construction. Two main roads are located about 800 m north and southeast of the sampling site and a motorway lies at a distance of 1.7

km east of the site. Most of the nearby land is covered by woods and agricultural fields. The station is also influenced by regional emissions from Paris and its surroundings (Ile-de-France, more than 12 million inhabitants). In spring and autumn,



the predominant wind direction is north-northeast transporting polluted air from Paris area while in winter and summer, the wind mainly blows from the northwest with relatively clean air from the ocean and less densely populated regions. Pal and Haeffelin (2015) showed that over the five years of their study of the planetary boundary layer (PBL) height at a location a few kilometers away from Saclay, the nocturnal PBL was above 100 m the sampling height of SAC tower, meaning we are

sampling within the nocturnal PBL most of the time.

Routine radon monitoring at SAC is conducted at 2 m, 50 m and 100 m above ground level (agl), while GHG are sampled at 15 m, 60 m and 100 m agl. Various meteorological measurements are available at 0.1, 1.5, 60 and 100 m agl. The above data have been measured since 2015 for GHG and meteorological data and since the end of 2016 for 100 m radon measurements. On the ICOS Carbon portal, GHG are available from May 2017 (date of ICOS fully compliant data) and radon from December

2020. The RTM has previously been applied at the nearby site of Gif-sur-Yvette, 2 km west of SAC (Messager, 2007; Yver et al., 2009; Lopez et al., 2012; Belviso et al., 2013, 2020). Yver et al. (2009) summarized the radon flux estimates before 2009 made with flux chambers and showed that they were ranging 42 – 66 +/- 22 Bq $m^{-2}h^{-1}$ with an average of 52 Bq $m^{-2}h^{-1}$. In summer 2013, additional measurements were done and used to assess Karstens et al. (2015) radon exhalation rate map (Schwingshackl, 2013). The values found for SAC were 18 – 54 Bq $m^{-2}h^{-1}$ for observations and models. Footprints for

the site using the Stochastic Time-Inverted Lagrangian Transport model (STILT) (Lin et al., 2003) are available on the ICOS Carbon Portal (https://stilt.icos-cp.eu/viewer/) from 2014 until the end of 2022.

$CO_2$, $CH_4$, CO and $N_2O$ are measured with cavity ring-down spectrometry analyzers from Picarro, Inc. In normal operation, one analyzer measures $CO_2$, $CH_4$ and CO (G2401 model) continuously at 100 m while two analyzers measure $CO_2$, $CH_4$, CO and $N_2O$ (G2401 and G5310 models) sequentially at 15, 60 and 100 m spending 10 minutes per level. This allows us to use

data average by thirty minutes to match the radon analyzer measurement frequency even for the $N_2O$ data in the rest of the study. Air is dried using nafion membranes as in Welp et al. (2012), target gases are measured on a daily basis and calibration gases on a monthly basis. The measurement repeatability is estimated with the regular analysis of a target gas (Kwok et al., 2015), which at Saclay can be rounded at about 0.03 ppm for $CO_2$, 0.1 ppb for $CH_4$, 0.3 ppb before 2019 then 0.03 ppb for CO (due to instrument change) and 0.05 ppb for $N_2O$. Systematic biases do no add uncertainty to the RTM as we are looking

at mixing ratio differences. Using a nafion also reduces strongly any diurnal variations and thus any potential bias due to the water vapor influence. Thus, the uncertainties on the measurements are about 0.3%, 0.3%, 0.6% then 0.06% and 5% of the diurnal cycle amplitude for $CO_2$, $CH_4$, CO and $N_2O$ respectively for the measurements at 100 m at Saclay. Radon is measured with a 1500 L ANSTO analyzer with a data every 30 minutes and its uncertainty is around 10% as described in Whittlestone and Zahorowski (1998); Grossi et al. (2019). For this study, we use the GHG and radon data measured at 100 m.

Figure 1 presents the mixing ratios and wind roses for $CO_2$, $CH_4$, CO and $N_2O$ at 100 m over the 2017-2022 period. The main wind direction is south-west in winter and autumn while in spring, wind comes also often from the north-west. In summer, wind covers almost the whole quadrant except south and east. For any season, the elevated concentration of $CO_2$, $CH_4$, CO and $N_2O$ are found in the north-west with the Paris area and further away the urbanised regions of Germany, Netherlands and Belgium.





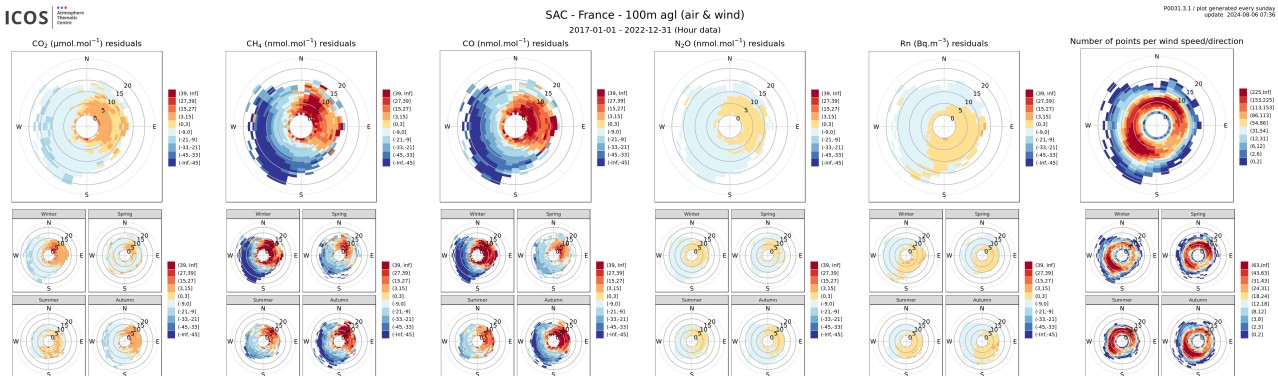

**Figure 1.** Mixing ratio and wind roses for $CO_2$, $CH_4$, CO, $N_2O$ and $^{222}Rn$ over the period 2017-2022. The residuals are calculated by subtracting the function fitted using the CCGCRV code from the data (Thoning et al., 1989). The top panels show the average residuals for the whole period while the lower panels are separated by season. For the GHGs, the radial axe shows the absolute number of data for each wind direction and the color scales shows the residual intensity while for the windrose, the axial axe represents the wind speeds and the color scale their frequency.

## 2.2 The Radon Tracer Method

In this study, we are focusing on the nocturnal accumulation RTM. The principle is based on the assumption of a constant flux $J_{gas}$ in a well-mixed layer of height $H$ during a nocturnal time window (8 to 10 hours window), thus we can write the temporal variation of its concentration as:

$$\frac{\Delta C_{gas}}{\Delta t} = \frac{J_{gas}}{H} \tag{1}$$

The same can be written for radon with an additional radioactive decay term.

$$\frac{\Delta C_{Rn}}{\Delta t} = \frac{J_{Rn}}{H} - \lambda_{Rn} C_{Rn} \tag{2}$$

If we combine equations 1 and 2 and we consider that for co-located measurement the height of the boundary layer is the same, we obtain:

$$J_{gas} = J_{Rn} \frac{\frac{\Delta C_{gas}}{\Delta t}}{\frac{\Delta C_{Rn}}{\Delta t}} \left(1 + \frac{\lambda_{Rn} C_{Rn}}{\frac{\Delta C_{Rn}}{\Delta t}}\right)^{-1} \tag{3}$$

which, for $\lambda_{Rn} C_{Rn} \ll \frac{\Delta C_{Rn}}{\Delta t}$ , simplifies to

$$J_{gas} = J_{Rn} \frac{\Delta C_{gas}}{\Delta C_{Rn}} \left(1 - \frac{\lambda_{Rn} C_{Rn}}{\frac{\Delta C_{Rn}}{\Delta t}}\right) \tag{4}$$





$J_{Rn}$ is the $^{222}$Rn flux over the region of influence, $\frac{\Delta C_{gas}}{\Delta C_{Rn}}$ is the slope of the linear regression of observations between the gases. Both mixing height and net surface flux of the catchment area are averaged for the observation period, and $(1 - \frac{\lambda_{Rn} C_{Rn}}{\frac{\Delta C_{Rn}}{\Delta t}})$ is the factor used to correct for $^{222}$Rn radioactive decay. As observations for greenhouse gases are usually reported as mixing ratios, it is necessary to convert them in concentrations before applying the RTM.

In this approach, the gas fluxes are considered similarly distributed in space and time, with no mixing of air from the free troposphere. The boundary layer height and the gas fluxes are assumed to remain constant during each nocturnal event. Figure 2 presents an example at Saclay, where the greenhouse gases and radon show the same behaviour and are thus good candidate to apply the RTM. During the first night, $R^2$ equals 0.95, 0.91, 0.91 and 0.94 for $CO_2$, $CH_4$, CO and $N_2O$ versus $^{222}$Rn respectively. On the second night, $R^2$ equals 0.94 and 0.87 for $CO_2$ and $N_2O$ versus $^{222}$Rn respectively. $CH_4$ and CO are not
correlated with $^{222}$Rn.

When we combine the RTM with air particle backtrajectories, we do not assume a regular region of influence to the radon concentration, but we consider that the influence of each grid cell around the station depends on the residence time of the incoming air over that grid cell (footprint). Hence, the radon flux $J_{Rn}$ is calculated weighting the radon flux of each gridcell by a sensitivity value (source-receptor matrix) obtained with the backtrajectory model (Seibert and Frank, 2004). More details on
this approach are described in Grossi et al. (2018).

## 2.3 The RTM software

An interactive tool to apply the RTM to estimate GHG fluxes from ICOS atmospheric concentration measurement was developed. The code is written in Python and is hosted on the ICOS Carbon Portal (CP) JupyterLab. It thus takes advantage of the ICOS CP Python package to access ICOS site data and already calculated footprints.

By default, it uses the footprints already calculated by the Lagrangian model STILT as configured on the CP (available for all ICOS sites and more for at least 2018 to 2022 https://stilt.icos-cp.eu/viewer/, last access: 03 October 2024). The STILT footprints are available every 3 hours and cover the region 33°S–73°N, 15°W–35°E with a resolution of 1/12° by 1/8°, approximately 10 km x 10 km. The STILT model is forced with the European Centre for Medium-Range Weather Forecast (ECMWF) Integrated Forecasting System (IFS) operational analysis. As these footprints are initially calculated for $CO_2$, no term for the
radon radioactive decay has been added.

The radon exhalation maps used are the two new maps developed in the project 19ENV01 traceRadon (using new input data sets such as soil uranium content and physical soil properties and either the reanalysed moisture data from ERA5-Land (Muñoz Sabater, 2019) or GLDAS-Noah2.1 (Beaudoing and Rodell, 2020)) with a value per day and available from 2017 to 2022. Their resolution is 0.05°x0.05° approximately 5.5 km latitude x 3.7 km longitude. All maps can be downloaded from the
ICOS CP (Karstens and Levin, 2023, 2024).

The radon exhalation maps and the footprints use different grids. Therefore, when combined, the radon exhalation maps are regridded to match the footprints.

The site to study can be chosen from the list available on the CP. The RTM can be applied to different species when data are available ($CO_2$, $CH_4$, $N_2O$ and CO). Then, either it extracts the data from the CP NRT hourly data or if the user has an access



**Figure 2.** Two days in May 2021 showing $CO_2$, $CH_4$, CO, $N_2O$ and $^{222}$Rn mixing ratios over time and their correlation. During the first night, $R^2$ equals 0.95, 0.91, 0.91 and 0.94 for $CO_2$, $CH_4$, CO and $N_2O$ versus $^{222}$Rn respectively. On the second night, $R^2$ equals 0.94 and 0.87 for $CO_2$ and $N_2O$ versus $^{222}$Rn respectively. $CH_4$ and CO are not correlated with $^{222}$Rn.



to the ICOS Atmosphere Thematic Center database with extraction rights for this site, data with a shorter timestep (minute data) can be extracted directly from the ICOS database and averaged on a 30 minute window to match the highest resolution for the radon data.

By default, the code applies the RTM equation for the data between 21:00 to 06:00 UTC, which is a suitable window for
central Europe where most of the ICOS stations are located, but this window can be modified to fit with other latitudes or longitudes for example to accommodate the shorter summer nights in northern Europe. The length of the window can be modified as well, for example to reproduce the tests from Levin et al. (2021). We apply an orthogonal distance regression using the SciPy.odr package.

No other criteria are applied initially but the correlation coefficient, the error in the linear regression, the number of data
points and hours available for the calculation, the radon accumulation level and whether the radon rise stopped before 08:00 UTC are recorded so the data can be filtered in a second step.

## 3   Sensitivity study of the RTM

### 3.1   Run description

For the sensitivity study, we added the possibility of using Karstens et al. (2015) exhalation map and atmospheric radon activity
measurements from csv files. For the ANSTO detector, there is a measurement response time to consider, due to their design (a combined influence of their thoron delay volume, large measurement volume, and gross alpha counting approach). For an optimal utilisation of radon measurements, a standardized protocol for data processing is required. This is not done yet fully in any ICOS radon data treatment chain though the first step of temperature and pressure normalization is now applied. For this work, we have used a radon dataset in a csv file derived from a standardized procedure, which is applicable to observations
made by any similar (ANSTO made) radon monitoring system. The procedure to obtain the best estimate of atmospheric radon concentration (final product data) involves the traceability and the post-processing of radon data, which includes the crucial deconvolution routine (step to correct for the instrument response time) as well as correction for standard temperature and pressure (STP). The details of the processing are available in Kikaj et al. (2024). The data that have been processed through this way are hereafter referred as standardized.

We also added the possibility to use footprints from another model. Each model grid has to be tailored to the footprint grid size. The FLEXPART-WRF model version 3.3.2 (Brioude et al., 2013) run at the Universitat Politècnica de Catalunya (UPC, Spain), is used here. This FLEXPART model uses ECMWF ERA5 meteorological files as inputs for its backtrajectory calculations. This model was used with an output resolution of 0.05 degrees in order to fit with the new ERA5-Land and GLDAS-Noah2.1 radon maps. The backtrajectories were calculated for a 24h window time and assuming the 0-100 m layer as
the footprint layer. For the Saclay site, the spatial domain used was 42.9°–54.5° in latitude and -6°–16.2° in longitude.

For the RTM runs (see Table 1), we used the three different radon exhalation maps available (called hereafter InGOS (climatology based on Karstens et al. (2015)), traceRadon_ERA5, traceRadon_Noah), two models (CP-STILT, WRF-FLEXPART) and two types of data (with and without standardization). By default, the two models do not compute the radon decay term.



**Table 1.** RTM runs for the sensitivity tests

| Run number | Run name | Model | Radon map | Radon data | Rn decay |
|---|---|---|---|---|---|
| 1 | STILT_InGOS_raw | STILT | InGOS | calibrated only ('raw') | No |
| 2 | STILT_InGOS_standard | STILT | InGOS | Standardized | No |
| 3 | STILT_ERA5_standard | STILT | traceRadon-ERA5 | Standardized | No |
| 4 | STILT_Noah_standard | STILT | traceRadon-Noah | Standardized | No |
| 5 | FLEXPART_ERA5_standard | FLEXPART | traceRadon-ERA5 | Standardized | No |
| 6 | FLEXPART_ERA5_standard_decay | FLEXPART | traceRadon-ERA5 | Standardized | Yes |

It is applied as a fixed term in the equation (3) as in Schmidt et al. (2001). For the last run, however, WRF-FLEXPART was modified to apply the Radon decay at each time step so the footprint already accounts for it. Not all combinations are tested but all runs can go in pairs, with only one change from one to the other. Two months were chosen: February 2019 and August 2019 to observe the seasonal influence and as months with a good data coverage for $CH_4$.

The setup for STILT_InGOS_raw and STILT_InGOS_standard were identical, except that radon data from the 1500L ANSTO monitor used was either the calibrated detector output or the standardized atmospheric radon concentration. This was done to study the impact of not using the standardization on the efficiency of the RTM application. STILT_ERA5_standard and STILT_Noah_standard were carried out using footprints calculated with the same CP-STILT model configuration and the same atmospheric concentration radon and GHG data. In this case, the radon flux maps traceRadon-ERA5 and traceR-

adon_Noah were used to evaluate how radon fluxes calculated using different soil moisture reanalysis data could influence the final results. FLEXPART_ERA5_standard used the same configuration as STILT_ERA5_standard, but with the FLEXPART-WRF based footprints which were calculated in the UPC cluster. Finally, STILT_ERA5_standard_decay was the same as STILT_ERA5_standard with the radon decay directly included in the footprint calculations.

    Radon fluxes were applied in different ways for each night during the two months:

1. constant radon flux value over the area of interest (52 Bq m$^{-2}$h$^{-1}$);

    2. radon flux values obtained with the available radon flux maps (InGOS, traceRadon_ERA5 and traceRadon_Noah) in the gridcell including the station. In the case of the InGOS map only one value per month was available while daily mean values are available for the two new traceRadon maps;

    3. radon fluxes values obtained coupling the previous radon flux maps with the atmospheric transport model (ATM) based

footprints for the studied night.

Methane fluxes within this study were calculated for every day during the months of February 2019 and August 2019 using at least two hours (or four datapoints) in the linear correlation between radon and $CH_4$.





The linear fits calculated between nocturnal radon and $CH_4$ data at the Saclay stations were then filtered to retain only the meaningful events using the following criteria: $R^2$ >0.6; error on the slope <50 %; radon increase over the night >1 Bq m$^{-3}$.

## 3.2 RTM sensitivity evaluation

Figure 3 shows the daily radon flux for February 2019 at the top and August 2019 at the bottom for the different runs and
three different fluxes. On the left, the fixed flux from the literature is displayed. In the middle, the fluxes are derived only from the station pixel of the different exhalation maps. On the right, the radon fluxes come from the combination of the exhalation maps and the nighttime footprint. For each panel, only the runs leading to different results are show, e.g. for the left panel, only STILT_InGOS_standard is shown as model and $^{222}$Rn exhalation map are not used with the fixed flux from the literature. Results show that $^{222}$Rn fluxes in winter are generally lower than those in summer, as it was expected from the
literature (Stranden et al., 1984). Indeed, in summer, the lower water content in the soil during this drier period leads to higher fluxes than in winter and spring, when the soil humidity can prevent the radon to exhale. In the middle panels, daily radon fluxes based on GLDAS-Noah reanalysis (STILT_Noah_standard) for this station pixel and period of time, display higher values than the ones calculated using ERA5-Land data (STILT_ERA5_standard). Specifically daily fluxes vary between 12 and 27 Bq m$^{-2}$h$^{-1}$ for STILT_ERA5_standard and in the range of 49 to 60 Bq m$^{-2}$h$^{-1}$ for STILT_Noah_standard, while
STILT_InGOS_standard is at 24 Bq m$^{-2}$h$^{-1}$ in February 2019. In August 2019, they vary between 68 and 86 Bq Bq m$^{-2}$h$^{-1}$ for STILT_ERA5_standard, between 105 and 112 Bq m$^{-2}$h$^{-1}$ for STILT_Noah_standard and STILT_InGOS_standard is at 55 Bq m$^{-2}$h$^{-1}$.

In the right panels, radon fluxes calculated using radon flux maps and ATM footprints show as expected a different variability, but the range is in the same order of magnitude. In February, the fluxes vary between 24 and 38 Bq m$^{-2}$h$^{-1}$ for
STILT_InGOS_standard, 15 and 42 Bq m$^{-2}$h$^{-1}$ for STILT_ERA5_standard, 44 and 71 Bq m$^{-2}$h$^{-1}$ for STILT_Noah_standard, and 15 and 38 Bq m$^{-2}$h$^{-1}$ for FLEXPART_ERA5_standard. In August, the fluxes vary between 26 and 60 Bq m$^{-2}$h$^{-1}$ for STILT_InGOS_standard, 31 and 88 Bq m$^{-2}$h$^{-1}$ for STILT_ERA5_standard, 44 and 119 Bq m$^{-2}$h$^{-1}$ for STILT_Noah_standard and 30 and 115 Bq m$^{-2}$h$^{-1}$ for FLEXPART_ERA5_standard. We do not see significant differences between FLEXPART_ERA5_standard and FLEXPART_ERA5_standard_decay showing that the approximation usually used for the radon decay is coherent with the
model results and can be used without degrading the results when the ATM footprint does not include the decay.

As shown in Figure 4, in winter, the radon exhalation rate is driving the difference between the different tests. Between STILT_InGOS_standard and STILT_Noah_standard, there is a 97% difference while between STILT_InGOS_standard and the other runs, the difference is below 12%. In summer, this difference is seen when using only the exhalation map pixel value but when using the footprint, the variability of all runs compared to STILT_ERA5_standard is between 6% and 75%.
STILT_Noah_standard still shows the higher difference but the FLEXPART model leads to difference almost as high using the lower radon exhalation rate as in STILT_ERA5_standard.

As can be expected, the variability on the radon fluxes is seen as well on the $CH_4$ fluxes (see Figure 5), reflecting that the day to day variation is due not only to the emission variations but also to the different areas that are sampled. Despite the fact that the standardized dataset allows to allocate the right sampling time for the radon measurement and thus when the two





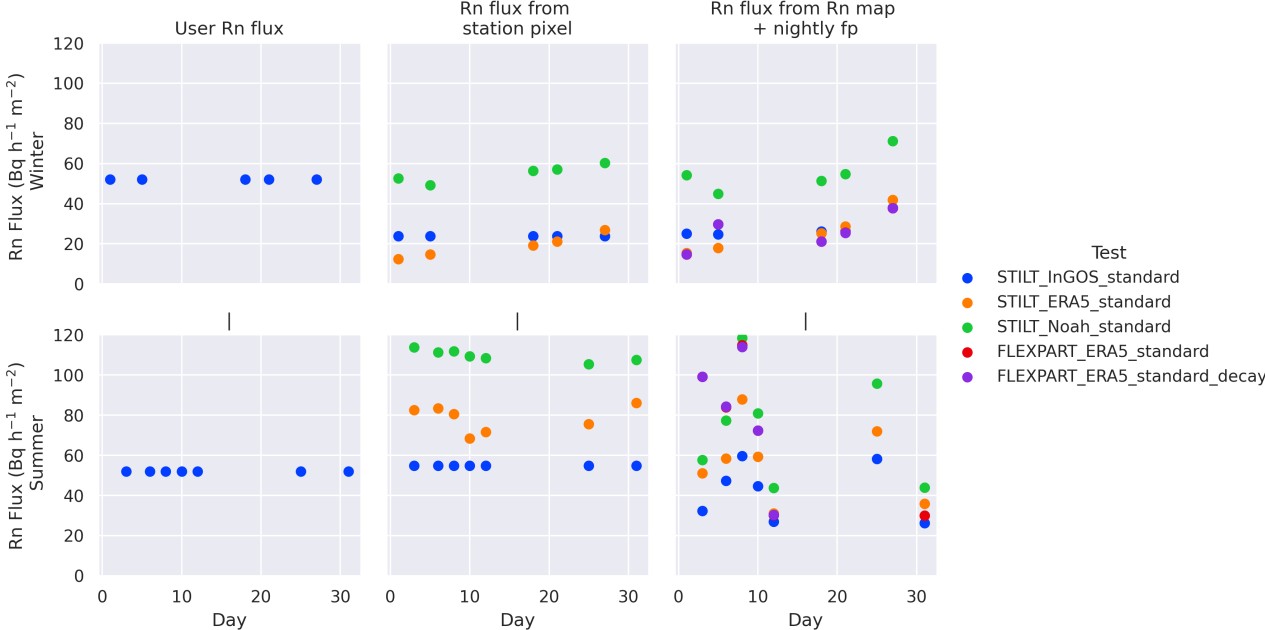

**Figure 3.** $^{222}$Rn flux in February (top) and August (bottom) 2019 for the sensitivity tests. On the left, the fixed flux from the literature is displayed. In the middle, the fluxes are derived only from the station pixel of the different exhalation maps. On the right, the radon fluxes come from the combination of the exhalation maps and the nighttime footprint. The colored dots represent the fluxes for the different runs. For each panel, only the runs leading to different results are shown for clarity.

gases are influenced by the same air masses their correlation should be better than when the data are not correcting and lagging behind, no significant differences is found between STILT_InGOS_raw and STILT_InGOS_standard in term of numbers of events selected especially in summer. In February, 5 events are found when using the standardized dataset and 2 without. In August, 6 events are found when using the standardized datasets and 7 without. To investigate this more thouroughly, STILT_InGOS_raw and STILT_InGOS_standard are applied for the whole year of 2021 (see Figure 6). 86 events are found when using the standardized datasets and 57 without. 48 events are found on the same days. The average $R^2$ are similar at 0.82 for both. The standardization nearly doubles the number of selected events and the difference between the resulting fluxes using the standardized method or not is about 20% with lower fluxes for the standardized method. On the whole year, we can see that not correcting the data for the response time leads to an underselection of events (poorer correlation) and to higher fluxes (smoothing effect of the large volume).

This study highlights the importance of using standardized radon activity data as well as the impact of the radon exhalation rate. The transport models used here have a lower influence while using a simplified decay term compared to have this term included in the model leads to insignificant differences.





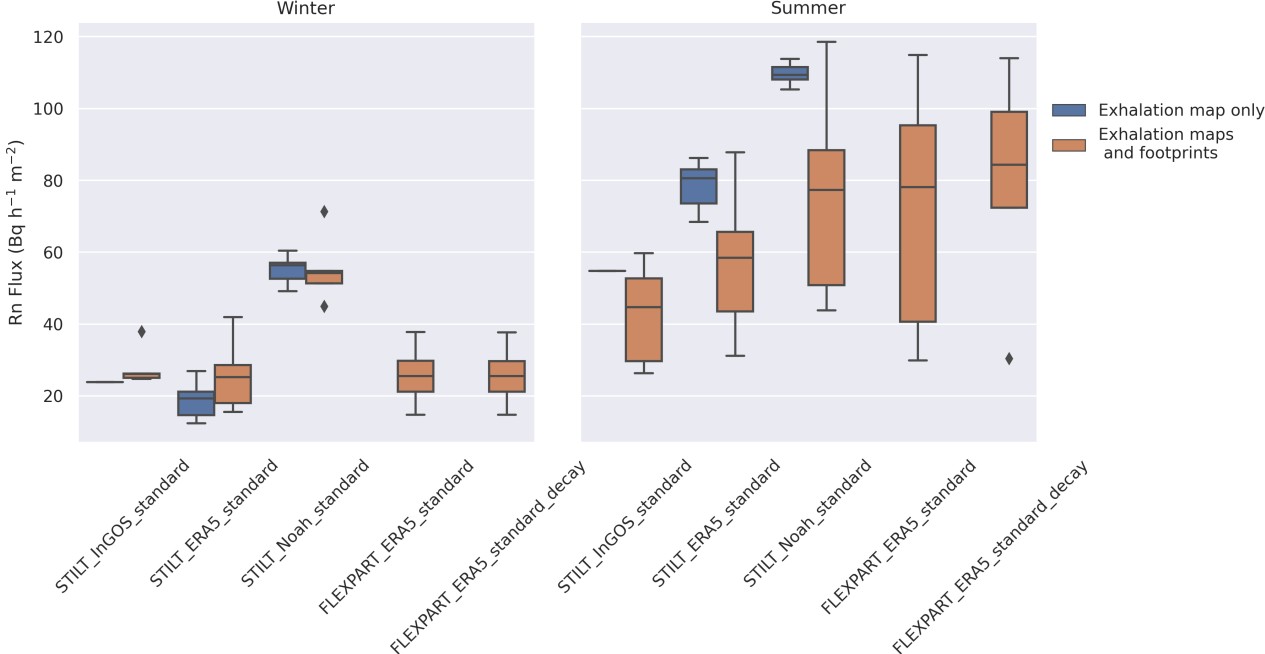

**Figure 4.** $^{222}$Rn mean flux in February (left) and August (right) 2019 for the sensitivity tests when either using only the exhalation rate maps or when using both the maps and the footprints

## 4 Six years of CO$_2$, CH$_4$, CO and N$_2$O flux estimates

Following the findings from the sensitivity study, for Saclay, we used standardized radon data, radon exhalation rate coming from the map using GLDAS-Noahv2.1 combined with the STILT footprint of each night (an aggregate of 3 footprints from 21:00 UTC, 00:00 UTC and 03:00 UTC). We combined them with CO$_2$, CH$_4$, CO and N$_2$O data from the ICOS database
5    at a 30 minute interval between 21:00 and 06:00 UTC. We chose to use the map using GLDAS-Noahv2.1 as Karstens and Levin (2024) showed that for Saclay, STILT_Noah_standard using the GLDAS-Noahv2.1 moisture parametrization and STILT model was exhibiting the smallest differences compared to measured data.

### 4.1 Radon fluxes

Figure 7 shows the radon exhalation rate from 2017 to 2023 from the exhalation map combined with the night footprints
10   compared to the exhalation rate from the same map but using only the station pixel or to a fixed value from the literature (Yver et al., 2009). Using the exhalation map allows to follow the seasonal pattern of the radon exhalation rate driven by humidity (Stranden et al., 1984). Combined with the footprints, the average rate is lower for SAC, especially in winter and the seasonal cycle is less marked: the summer maximum is still visible but lower values are found throughout the year. It highlights the importance of well defining the influence zone as the uncertainties on one pixel are larger than on the whole map. The average




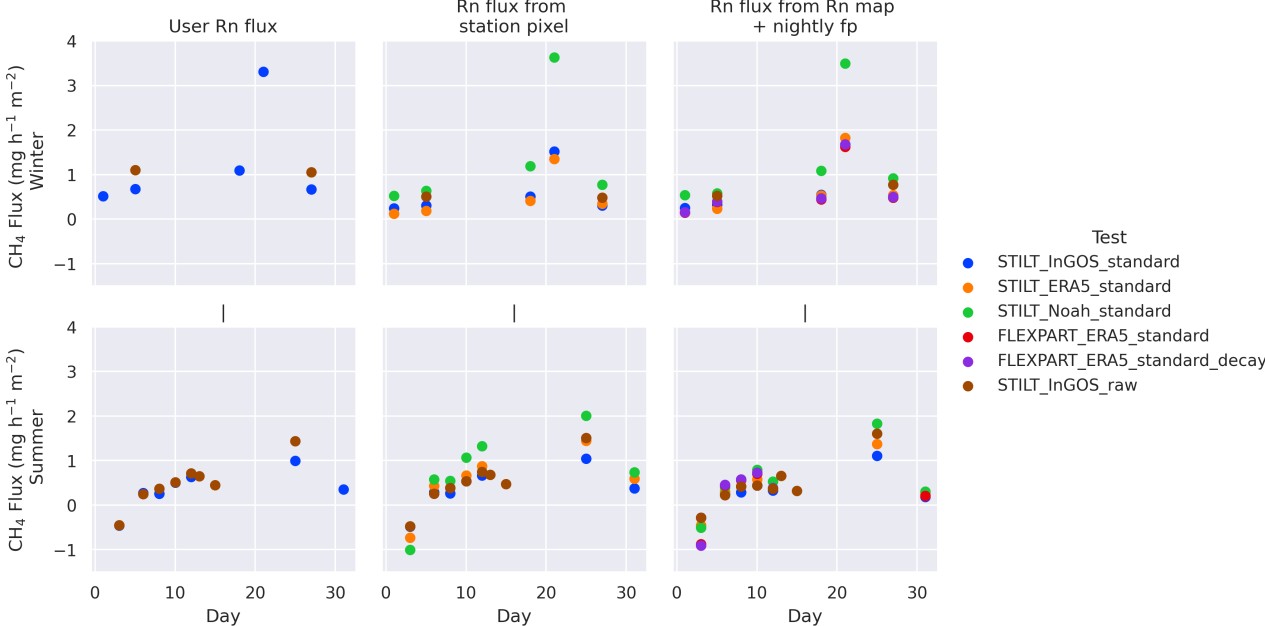

**Figure 5.** $CH_4$ flux in February (top) and August (bottom) 2019 for the sensitivity tests. On the left, the fixed flux from the literature is displayed. In the middle, the fluxes are derived only from the station pixel of the different exhalation maps. On the right, the radon fluxes come from the combination of the exhalation maps and the nighttime footprint. The colored dots represent the fluxes for the different runs. For each panel, only the runs leading to different results are shown for clarity.

winter and summer footprint calculated using the RTM selected nights for 2021 are shown in Figure 8 together with the radon exhalation averaged over the same period. We clearly see the increased radon exhalation rate in summer compared to winter in the right panels. The difference in the footprint is less marked but in winter a narrow west sector and the south-east sector do not influence the station while in summer, the station is influenced from all sectors with a reduced influence in the south. Over

5 the six years, the average radon flux rate estimated from the footprint analysis applied to the GLDAS-Noah map reaches 52 Bq $m^{-2}h^{-1}$ which corresponds to the literature value that has been used previously. The minimum is 14 Bq $m^{-2}h^{-1}$ and the maximum 118 Bq $m^{-2}h^{-1}$ leading to a larger amplitude than previously applied at Saclay (25% Yver et al. (2009)). We also observe an annual variability with the lowest values in 2017 and the highest in 2019 (see Figure 7). In 2017 and in 2021, the amplitude is smaller with less high values in summer than the other years. This is linked with the moisture reproduced by the

10 GLDAS-Noah model, as can be seen in Figure 9 where 2017 and 2021 show higher moisture content in summer than during the other years. The value for the station pixel is much higher at 72 Bq $m^{-2}h^{-1}$ with the same maximum but higher minimas than when using the footprints as well.







**Figure 6.** CH$_4$ flux in 2021 using STILT_ERA5_raw or STILT_ERA5_standard




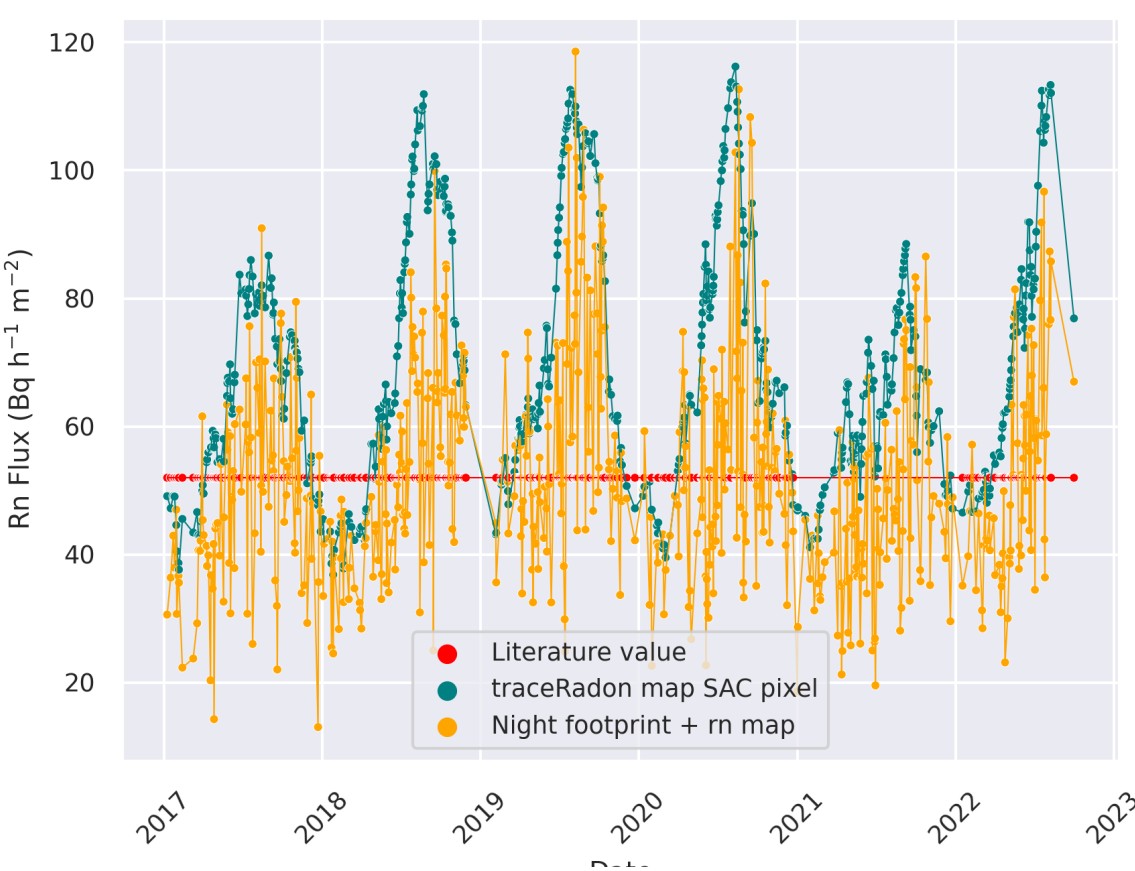

**Figure 7.** $^{222}$Rn flux from 2017 to 2023 for $CO_2$ selected nights with the different ways of calculating the flux: in red, the user value from the literature, in blue, the value from the traceRadon exhalation rate map for the SAC pixel and in yellow, the value from combining the traceRadon map with the STILT footprints for each night.



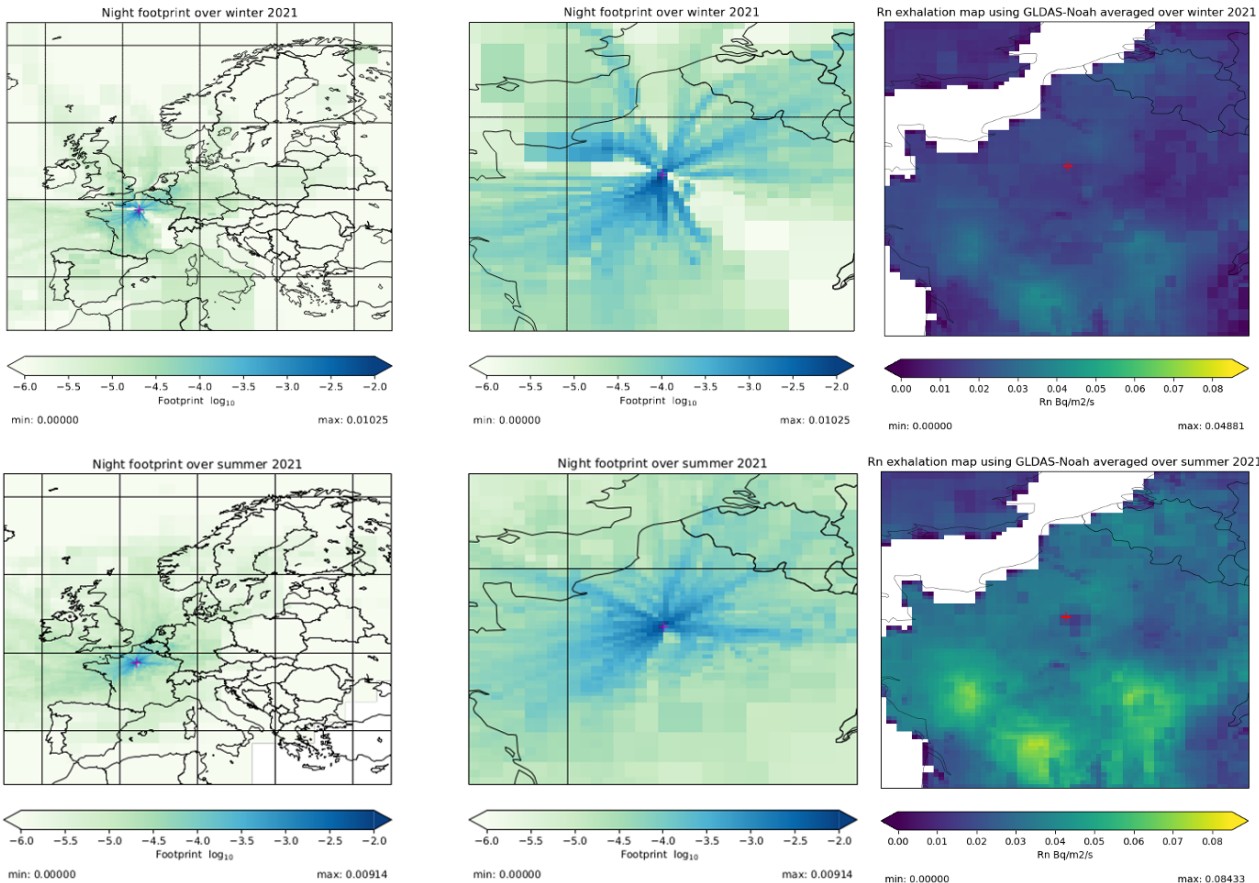

**Figure 8.** Saclay STILT footprint averaged over the RTM selected days for CH$_4$ in winter and summer 2021 along with the traceRadon_Noah exhalation rate map for the same periods. Winter is on top and summer on the bottom. On the left panels, the footprints over the whole STILT domain is shown, in the middle panel, we show a zoom around SAC. The color scale represents the sensitivity of each pixel to the emissions reaching SAC. The exhalation rate maps are shown on the right panels.



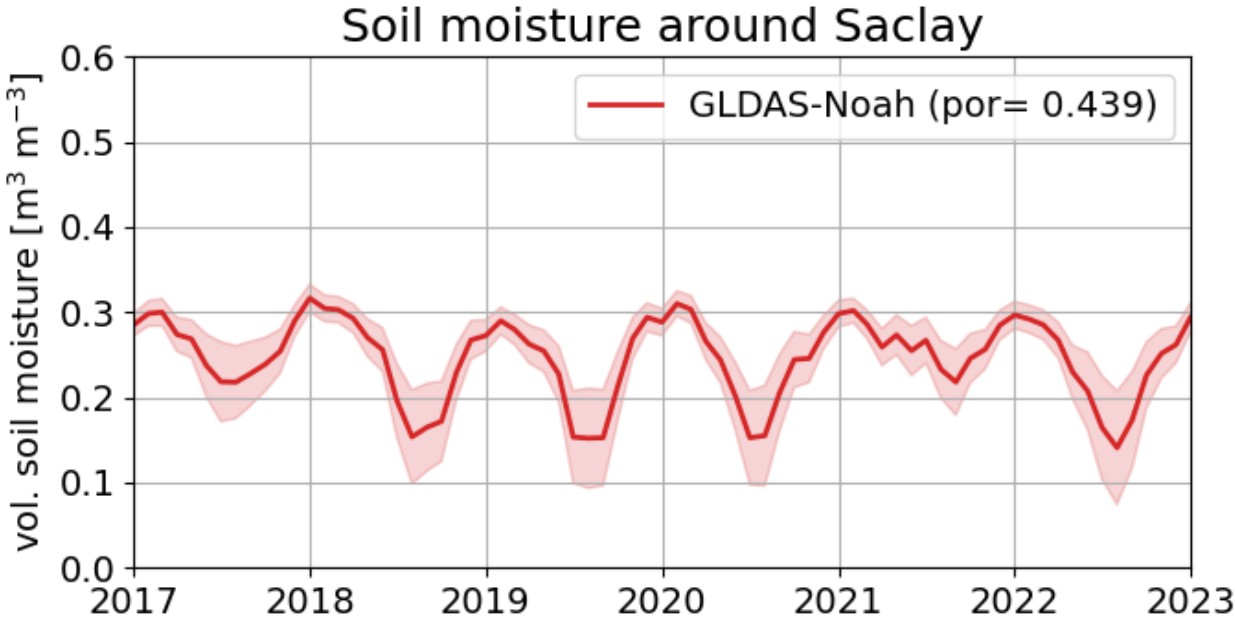

**Figure 9.** Soil moisture content for the pixel containing SAC as modeled in GLDAS-Noahv2.1 from 2017 to 2023

## 4.2 Bottom-up emissions

We use two different bottom-up inventories for comparison to the RTM derived GHG fluxes. The first one is EDGAR v8.0 (Crippa et al., 2022). These are gridded annual anthropogenic fluxes with a 0.1° resolution over the world for $CO_2$, $CH_4$ and $N_2O$. For $CO_2$, $CH_4$ and CO, we use the Netherlands Organisation for Applied Scientific Research (TNO) inventory (Kuenen

et al., 2014; Super et al., 2020) which provides anthropogenic fluxes with a 6 km resolution over Europe. Monthly, weekly and daily profiles can be added to the TNO emission map. For this work, we only apply the monthly profile to the TNO inventory to reflect the seasonal changes. The TNO daily profile shows no variations for $CH_4$ but a diminution of about 50% for the emissions between 21:00 and 06:00 UTC for $CO_2$. However, the air measured at midnight at Saclay travelled during the day and it is thus not straightforward to select which hours of the inventory should be favored. For this reason, we choose not to

apply the daily profile.

For biogenic $CO_2$ fluxes, we compare our results to simulations using the WRF-VPRM model with a 5 km resolution around the Ile-de-France area (Lian et al., 2019). WRF-VPRM winds were forced by the hourly reanalysis fields from the fifth generation of meteorological analysis of the ECMWF at 0.75°x0.75° resolution, respectively (ECMWF ERA5, Hersbach et al. (2020)). The VPRM model is forced by meteorological fields simulated by WRF and online-coupled to the atmospheric

transport. It uses vegetation indices derived from the 8d MODIS Surface Reflectance Product (MOD09A1) and four parameters optimized using data from the Integrated EU project "CarboEurope-IP" (https://www.copernicus.eu/en/carbo-europe-ip, last





access: 03 October 2024). VPRM uses a land cover derived from the 1 km global Synergetic Land Cover Product (SYNMAP; Jung et al. (2006)) reclassified into eight different vegetation classes (Ahmadov et al., 2007, 2009).

EDGAR, TNO and VPRM fluxes have been combined with the footprint of each selected event to calculate the fluxes comparable to the fluxes estimated with the RTM. In the rest of the text, we call these combinations EDGARf, TNOf and VPRMf.

## 4.3 Uncertainties

To assess the validity of the fluxes deduced from the RTM and provided by inventories, we need to estimate their uncertainties. Here, the uncertainty of one RTM-based GHG flux estimate is the combination of the measurement uncertainty as described in section 2.1, the linear regression uncertainty on the slope and the radon flux uncertainty.

For the four species, the uncertainties on the linear regression is in average 14%, which combined with the measurement uncertainties yields a maximum uncertainty of 15% on the slope. The full uncertainty depends critically on the uncertainty in the modelled radon fluxes. Based on the underlying radon flux model and the assumed uncertainties in the input parameters, Karstens et al. (2015) estimated the uncertainty of modelled fluxes for individual pixels to about 50% and to approximately 30% when averaged over a larger area like in this application. Additionally, the radon flux maps show large differences depending on the soil moisture reanalysis that were used in the model (see Fig. 3 middle panel). Calibration of the radon flux map in the footprint region with long term measurements at the station could help to reduce this systematic uncertainty, while retaining information on temporal and spatial variability (Levin et al., 2021). We estimate here the RTM maximum total uncertainty at 35%. It is worth to note that the systematic errors or biases that are seen here are of less impact when studying the long term trend of emissions. In our study, we estimate 6 years of data and thus can begin to draw observations about potential trends.

EDGAR's uncertainties are discussed in Solazzo et al. (2021). Figure 12 of that paper summarizes well the uncertainties. For Europe, the uncertainties for $N_2O$ are 70%, 9% for $CH_4$ and 21% for $CO_2$.

TNO's uncertainties are discussed in Super et al. (2020) but only for $CO_2$ and CO. For the domain, a European region centred over the Netherlands and Germany, the uncertainties for the total emissions are 1% for $CO_2$ and 6% for CO. The largest contributor of uncertainties for $CO_2$ is the public power category while it is the stationary combustion for CO. The uncertainty increases up to 40% for $CO_2$ and 70% for CO if looking at specific grid cells.

## 4.4 $CH_4$ fluxes

Figure 10 shows the fluxes of $CH_4$ calculated using the STILT_Noah_standard configuration with the footprints. The figure also includes the bottom-up flux estimate for each footprint using EDGAR and TNO. The averages over 2017-2022 are shown in Table 2.

First, the RTM fluxes, EDGARf and TNOf are in a relatively good agreement showing a similar variability visible both in the left panels. No trend is observed. Although we do not observe a clear seasonal variability, January and February are usually showing lower emissions than the other months and less events in total. As EDGARf variability only comes from the footprints (annual inventory), this would be indicative of a sector with low emissions or no correlation sampled more often during these





**Figure 10.** The CH$_4$ fluxes (RTM), calculated using the STILT_Noah_standard configuration were compared with the combined EDGAR and TNO inventories using the same footprints for the period 2017-2023 (EDGARf and TNOf respectively). The top left panel shows all the selected data, the top right panel shows the correlation between the RTM fluxes and the inventory estimates. The bottom left plot shows the monthly average of the different estimates with the standard deviation as a shaded area while the bottom right plot shows the flux repartition over the windrose with the flux intensity as the color scale and its frequency on the axial axe.





**Table 2.** GHG fluxes (mg m$^{-2}$h$^{-1}$) averaged over the measurement period (2017-2022 or 2019-2022) for the RTM estimates fluxes using the STILT_Noah_standard configuration, for EDGARf, TNOf and VPRMf estimates. The second value represents the standard deviation over the whole period.

| Species | RTM | EDGARf | TNOf | VPRMf |
|---------|-----|--------|------|-------|
| CH$_4$ | 0.81 ± 0.66 | 0.63 ± 0.40 | 0.46 ± 0.24 | |
| CO$_2$ | 609 ± 402 | 142 ± 128 | 118 ± 98 | 89± 47 |
| N$_2$O | 0.063 ± 0.079 | 0.026 ± 0.01 | - | |
| CO | 1.04±1.80 | - | 0.53 ± 0.33 | |

months. This can be due to a boiler room located very close to the station, which signature can be seen when the wind is coming from the north-northeast. In case of the airmass coming over the boiler, the radon and methane would not present a correlation anymore. In January 2019, there was a lack of radon data for half of the month (16 days without data) which explains part of the gap on the bottom panel as only one event was found in January. However, in December 2018, there was no lack of data but no event found. For this month, the correlation was too low for 19 days and for the others, it was either the radon increase that was too low or the number of available hours. In November 2021, there is no lack of data but only one event selected. Here, as well, the correlation was too low for 19 days.

The RTM gives an average of 0.81 ± 0.66 mg CH$_4$ m$^{-2}$h$^{-1}$ which compares well with the EDGARf average of 0.63 ± 0.53 mg CH$_4$ m$^{-2}$h$^{-1}$ within the range of uncertainties. The TNOf value is almost a factor of two smaller than the RTM estimate with 0.46 ± 0.24 mg CH$_4$ m$^{-2}$h$^{-1}$. This is obvious in Figure 10 right top panel where we can see that all TNOf emissions are lower than the RTM. For EDGARf, in the lower range, the agreement is better but for the higher values, the RTM is systematically above EDGARf. As expected, the bottom right plot shows the highest emissions coming from the north-east, hence from the Paris area. Both EDGAR and TNO have agriculture and waste as the first most important sectors for Saclay average footprint, accounting for almost 80% of all the emissions with about the same share for the two sectors. The difference between the two inventories seems then to be in the total emissions and not in their sectorisation in this footprint.

Using the RTM, Levin et al. (2021) calculated an estimate for the Heidelberg region between 2015 and 2020 which reached an average of 0.8 mg CH$_4$ m$^{-2}$h$^{-1}$ while in Cabauw between 2016 and 2018 (Tong et al., 2023), the estimate reached 1.4 mg CH$_4$ m$^{-2}$h$^{-1}$ with the RTM or 1.48 CH$_4$ m$^{-2}$h$^{-1}$ with the vertical gradient method highlighting the importance of dairy farming in this region. At Gif-sur-Yvette, neighbouring Saclay, values of 0.8 mg CH$_4$ m$^{-2}$h$^{-1}$ were already found for the period 2002-2007 using the RTM and the boundary layer height (Messager, 2007).

### 4.5 N$_2$O fluxes

The fluxes of N$_2$O calculated using the STILT_Noah_standard configuration with the footprints are shown in Figure 11. As shown in the left panels, the RTM fluxes show a seasonal variability with a maximum around spring and summer while the EDGARf fluxes do not, reflecting the fact that it is a yearly estimate and that the seasonality is not driven by the variability





of air mass origins. No trend is observed. $N_2O$ RTM fluxes for 2019 and 2020 show the highest values in spring followed by a decrease over the rest of the year. In 2021 and 2022, the fluxes show spring high values as well but also in summer with higher values than the previous years in average. EDGARf shows much lower values than the RTM fluxes in average: 0.026 mg $N_2O$ m$^{-2}$h$^{-1}$ versus 0.063 mg $N_2O$ m$^{-2}$h$^{-1}$, being close to the lower values estimated with the RTM. Indeed, when

calculating the median, the difference decreases with 0.023 mg $N_2O$ m$^{-2}$h$^{-1}$ versus 0.037 mg $N_2O$ m$^{-2}$h$^{-1}$ for EDGARf and RTM respectively. The top right panel shows indeed a better correlation for the lower values and an underestimation for the higher ones. Agricultural soils are the main sector in EDGAR, accounting for more than 50% of the total but the spikes during fertilization episodes are not reported as it is a yearly estimate. The bottom right panel highlights this pattern, with most of the events located in the south-west where agriculture is more predominant than in the north-east where Paris and its suburban

areas lie.

   $N_2O$ fluxes were estimated at Cabauw as well (Tong et al., 2023) with an estimate of 0.046 mg $N_2O$ m$^{-2}$h$^{-1}$ with RTM and 0.068 mg $N_2O$ m$^{-2}$h$^{-1}$ with the vertical gradient method. At Gif-sur-Yvette, values of 0.068 mg $N_2O$ m$^{-2}$h$^{-1}$ were reported for the period 2002-2007 by Messager (2007) but within the range of 0.039 to 0.058 mg $N_2O$ m$^{-2}$h$^{-1}$ over the period 2002-2011 in Lopez et al. (2012). These values are of the same order of magnitude as the values found in this study. In studies measuring

fluxes directly in agricultural soils though soil chambers within the Ile-de-France region over several years (Colnenne-David et al., 2021; Garnier et al., 2024), maximum values of 1.4 mg $N_2O$ m$^{-2}$h$^{-1}$ and 2.3 mg $N_2O$ m$^{-2}$h$^{-1}$ were found and in general high values were distributed over spring and summer months. The average values reached 0.09 mg $N_2O$ m$^{-2}$h$^{-1}$ and 0.05 mg $N_2O$ m$^{-2}$h$^{-1}$ respectively. Garnier et al. (2024) also studied the factors influencing the $N_2O$ releases and found that rainfall (hence soil moisture) had the largest influence over $N_2O$ emissions, followed by daily maximum temperature.

## 4.6   CO fluxes

Figure 12 shows the fluxes of CO calculated using the STILT_Noah_standard configuration with the footprints. CO RTM and TNOf fluxes do not show a clear seasonal cycle or a trend over the period as shown in the left panels. The RTM fluxes reach an average of 1.04 mg CO m$^{-2}$h$^{-1}$. This is twice higher than the TNOf values for the same area. However, this difference is driven by a few high values and the median for the RTM fluxes is 0.55 mg CO m$^{-2}$h$^{-1}$ in very good agreement with TNOf

(0.44 mg CO m$^{-2}$h$^{-1}$). This pattern is seen in the top right panel where the values above 1 mg CO m$^{-2}$h$^{-1}$ are systematically underestimated by the inventory. For CO, more than 50% of the emissions come from the residential sector (other stationary combustion sources), then 20% from the road transport and 10% from the industry. Most of the fluxes are located in the south-west cadran, maybe indicative of a local source coming from the developement of the university campus close to the site. Messager (2007) found an average value of 1.46 mg CO m$^{-2}$h$^{-1}$ for Europe using measurement from Mace Head, Ireland

over the period 1996-2006 with a tendency to decrease over time.

## 4.7   $CO_2$ fluxes

Contrary to the other gases, $CO_2$ is also strongly influenced by biogenic emissions not represented in EDGAR or TNO. This is why we also used VPRM. Figure 13 shows the fluxes of $CO_2$ calculated using the STILT_Noah_standard configuration with





**Figure 11.** The $N_2O$ fluxes (RTM), calculated using the STILT_Noah_standard configuration were compared with the combined EDGAR inventory using the same footprints for the period 2019-2023 (EDGARf). The top left panel shows all the selected data, the top right panel shows the correlation between the RTM fluxes and the inventory estimates. The bottom left plot shows the monthly average of the different estimates with the standard deviation as a shaded area while the bottom right plot shows the flux repartition over the windrose with the flux intensity as the color scale and its frequency on the axial axe.





**Figure 12.** The CO fluxes (RTM), calculated using the STILT_Noah_standard configuration were compared with the combined TNO inventory using the same footprints for the period 2017-2023 (TNOf). The top left panel shows all the selected data, the top right panel shows the correlation between the RTM fluxes and the inventory estimates. The bottom left plot shows the monthly average of the different estimates with the standard deviation as a shaded area while the bottom right plot shows the flux repartition over the windrose with the flux intensity as the color scale and its frequency on the axial axe.



the footprints. No trend is observed. Looking at the RTM calculated fluxes, we observe a seasonal pattern with a minimum in winter and a maximum in summer mostly visible on the top left panel. Indeed, we are looking here at nocturnal fluxes without photosynthesis only respiration. The respiration as shown in Belviso et al. (2022), Figure 4, modeled for Saclay over 2015 to 2021, presents a seasonal cycle with a maximum in summer and minimum in winter, an average of 130 mg $CO_2$ m$^{-2}$h$^{-1}$ and

an amplitude of about 200 mg $CO_2$ m$^{-2}$h$^{-1}$. The amplitude of the RTM fluxes is in average 750 mg m$^{-2}$h$^{-1}$ so only a third would be accounted for the respiration. Several reasons can explain the difference, first the seasonal respiration in Belviso et al. (2022) is shown only for one grid pixel while in our study, we aggregate the fluxes for the station footprint, where there could be higher gradients of respiration. Secondly, there is of course the anthropogenic contribution of $CO_2$ as Saclay is a peri-urban site within the influence of the Paris region. The average estimate from EDGARf is 142 mg $CO_2$ m$^{-2}$h$^{-1}$ versus 609 mg

$CO_2$ m$^{-2}$h$^{-1}$ for the RTM. TNOf presents even lower values than EDGARf with an average of 118 mg $CO_2$ m$^{-2}$h$^{-1}$. On the figure, we see that EDGARf and VPRMf combined reproduce fairly well the baseline and winter fluxes estimated by the RTM. However, in summer, the RTM fluxes can be two to three times higher. This is highlighted in the top right panel where the underestimation is clear for all the bottom-up inventories. The lower right panel shows the higher values coming from the north-east like for $CH_4$.

The RTM values agrees with the value found for 2002-2007 in (Messager, 2007) of 545 mg $CO_2$ m$^{-2}$h$^{-1}$. For the two inventories, transportation, industry and stationary combustion are the three main sources with about the same proportions for both.

If we look at the $CO/CO_2$ ratio shown on Figure 14, we see that in summer, the ratio is below 1.5 showing a low influence from the anthropogenic sources and a higher one from the biogenic sources (Ammoura et al., 2014, 2016; Gamage et al., 2020).

This seems to point towards a misrepresentation of the biogenic fluxes from this VPRM version within our footprint. This is supported by Lian et al. (2023) where the authors show that the nighttime biosphere signals during the growing season are not well reproduced by the VPRM model for the Ile-de-France region where Saclay is located.

## 5 Conclusions

This study presents six years of $CO_2$, $CH_4$, CO and $N_2O$ fluxes at Saclay, France from 2017 to 2022. The fluxes have been

calculated using the radon tracer method. An interactive tool has been developped. The RTM has been combined with the latest radon flux maps from the traceRadon project and the STILT footprints as calculated in the ICOS Carbon Portal. A preliminary study focused on $CH_4$ and two months in 2019 helped define the best parameters to use for Saclay. To apply this tool to other stations, it would be potentially necessary to adjust the nocturnal window. An automatic selection of the sunset–sunrise period could be a development in the future.

In this study, we highlighted the importance of using radon standardized data as well as the impact of the radon exhalation rate maps that remains the main factor of uncertainties in the method despite recent improvements. We also showed that a simple radon decay correction is sufficient to yield accurate results.

In section 4, the method has been applied to all six years of data and four gases.





**Figure 13.** The $CO_2$ fluxes (RTM), calculated using the STILT_Noah_standard configuration were compared with the combined EDGAR, TNO and VPRM inventories using the same footprints for the period 2017-2023 (EDGARf, TNOf and VPRMf respectively). The top left panel shows all the selected data, the top right panel shows the correlation between the RTM fluxes and the inventory estimates. The bottom left plot shows the monthly average of the different estimates with the standard deviation as a shaded area while the bottom right plot shows the flux repartition over the windrose with the flux intensity as the color scale and its frequency on the axial axe.





**Figure 14.** CO/CO$_2$ ratio from 2017 to 2023 calculated over the common days where both CO and CO$_2$ fluxes were estimated.

We found that the average RTM estimates and their variability are $609 \pm 402$ mg m$^{-2}$h$^{-1}$, $0.81 \pm 0.66$ mg m$^{-2}$h$^{-1}$, $1.04 \pm 1.80$ mg m$^{-2}$h$^{-1}$ and $0.063 \pm 0.079$ mg m$^{-2}$h$^{-1}$ for CO$_2$, CH$_4$, CO and N$_2$O respectively.

CH$_4$, N$_2$O and CO are in fairly well agreement with the inventories, though with higher values. N$_2$O differences most probably come from the lack of seasonality in EDGAR inventory. CO$_2$ fluxes are about five times higher than anthropogenic and biogenic fluxes from EDGAR and VPRM combined. The differences mainly occur during summer, and the CO/CO$_2$ ratio points toward a misrepresentation of the biogenic fluxes at this time by the VPRM version used here.

The literature for either older measurement close to Saclay, or recent measurements in Germany and Netherlands agree fairly well with our findings notwithstanding the difference in the local environment like for methane in the Netherlands where dairy farming is much more prominent than at Saclay.





*Code and data availability.* Radon and GHG datasets, radon exhalation maps and STILT retrotrajectories are available on the ICOS Carbon Portal. The FLEXPART retrotrajectories used for the sensitivity study are available on demand. The RTM code is not directly available on the Carbon Portal but the notebook can be shared on demand.

*Author contributions.* C Yver-Kwok wrote the article, the RTM code and ran it. Edward Chung provided the radon deconvoluted data. Ute
5  Karstens provided the radon exhalation map and modeled radon concentration for Saclay. Roger Curcoll provided the FLEXPART runs. Jinghui Lian provided the VPRM runs. All authors contributed to the article correction and improvement.

*Competing interests.* The authors declare no competing interests.

*Acknowledgements.* This project 19ENV01 traceRadon has received funding from the EMPIR program co-financed by the Participating States and from the European Union's Horizon 2020 research and innovation program.



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
