# Peer review of "Measurement report: Eight years of greenhouse gas fluxes at Saclay, France, estimated with the Radon Tracer Method"

_EGUsphere, 2024_

## Author Comment (AC1)

**Six years of greenhouse gas fluxes at Saclay, France, estimated with the Radon Tracer Method**

Camille Yver-Kwok, Michel Ramonet, Léonard Rivier, Jinghui Lian, Claudia Grossi, Roger Curcoll, Dafina Kikaj, Edward Chung, and Ute Karstens

**Review 1**

https://doi.org/10.5194/egusphere-2024-3107-RC1

This paper used the Radon Tracer Method (RTM) to estimate greenhouse gas ($CO_2$, $CH_4$, $N_2O$) and CO fluxes at Saclay, France during the period of January 2017 – December 2022. The authors examined the sensitivity of the method to the use of different Radon exhalation maps. Radon exhalation maps from the 19ENV01 traceRadon project, STILT back trajectories from the ICOS Carbon Portal, estimates of radon activities and greenhouse gas data were then used to estimate surface emissions. They found that the estimated $CO_2$, $CH_4$, CO and $N_2O$ surface emissions were in good agreement with the literature and that $CH_4$, $N_2O$ and CO fluxes were also in fair agreement with inventories. The observation-based RTM method provides an independent approach (alternative to inverse modeling) to verify greenhouse gas fluxes, as demonstrated in this study. This reviewer's major concern is that the presentation of this paper needs improvement and in some places the texts are hard to understand (see examples below). Publication on ACP is recommended after serious editing and addressing the comments below.

*We thank the reviewer for his/her helpful comments about the form of the manuscript. Our answers are shown below each point and in italic.*

Abstract, Line 12: "$CH_4$, $N_2O$ and CO are also in fair agreement with the inventories, though with higher values" – do you actually mean "$CH_4$, $N_2O$ and CO fluxes"? "To our knowledge, this is the first study using the latest radon exhalation maps and standardized radon measurements to estimate $CO_2$, $CH_4$, CO and $N_2O$ surface emissions" - Is this for any site or for Saclay only? "These fluxes are in good agreement with the literature" – Could you cite the values from the literature for each species?

*To our knowledge, it is for any site. The latest maps have been used in Curcoll et al.,2024 but not in combination with the standardized radon measurements. We will add it and the literature values in the abstract.*

Page 3, Line 8: Kikaj et al. (2024) – when was this submitted? Not available to the reviewer.

*This paper was still in discussion at the time of the submission, it is now published :*
*https://doi.org/10.5194/amt-18-151-2025*

Page 3, Line 10: "the radon flux was considered homogeneous over time and space" – is this said for Paris or Europe? Probably this was an assumption made in the study of Yver et al. (2009)? "as it is now known that the radon fluxes varies on space and time" – it is long known (way before 2009) that the radon fluxes vary on space and time.

*It is indeed known before Yver et al., 2009 that the radon fluxes varies over time and space but before about that date we did not have access to spatialized maps, only individual measurements at different times and places. We will reformulate to clarify.*

Page 4, Line 4: "the nocturnal PBL was above 100m…." – I think you meant the nocturnal PBL height was above the 100 m sampling height of SAC tower.

*This will be reformulated for clarity.*

Figure 1 caption: what is the CCGCRV code?

*The CCGCRV code is a digital filtering curve fitting program developed by Kirk Thoning (Carbon Cycle Group, Earth System Research Laboratory (CCG/ESRL), NOAA, USA) in the late 1980s. It is first used in Thoning et al., 1989 and is available on*
*https://gml.noaa.gov/ccgg/mbl/crvfit/crvfit.html*

*We will clarify the text to add this explanation.*

Page 5, Line 11: Under which conditions will this (<<) be valid?

*For short-term variations of $C_{Rn}(t)$, eight hours in our study, we can assume that $\lambda_{Rn} C_{Rn} \ll \Delta C_{Rn}/\Delta t$, especially as in our study, we also apply a threshold on the radon increase to select events with a significant increase.*

*In Levin et al., 2021, the whole effect of the decay, estimated to be less than 10% is even neglected. Here, we apply the correction as defined in Schmidt et al.,2001.*

Page 10, Lines 9-11: It's well known that radon emissions under freezing temperatures in winter are much reduced. Is the higher soil humidity, which prevents the radon from exhaling, due to low temperature in winter?

*It is both due to a reduced evaporation and an increased amount of precipitation and condensation. It will be added in the text.*

Figure 3: "the fixed flux from the literature" --- which literature?

*We are referring here to the literature average established in Yver et al., 2009. We will clarify in the caption.*

Figure 5 caption: it's not clear whether "fluxes" are for Rn or CH4. Please clarify to avoid confusion.

*There is indeed some confusion in this caption. It will be clarified in the revised version.*

*"CH$_4$ flux in February (top) and August (bottom) 2019 for the sensitivity tests. On the left, the results from using a fixed radon flux from the literature (Yver et al., 2009) is displayed. In the middle, the methane fluxes come from the radon fluxes derived only from the station pixel of the different exhalation maps. On the right, the methane fluxes are derived from the radon fluxes calculated using the combination of the exhalation maps and the nighttime footprint. The colored dots represent the fluxes for the different runs. For each panel, only the runs leading to different results are shown for clarity."*

Figure 6: "CH4 2 Flux"?

*The typo will be corrected.*

Figure 8 caption: "On the left panels, …shown, in the middle panel, we show…" -  Editing is needed.

*We will rephrase for clarity.*

Figure 9: what is "por"?

*"por" stands for porosity. We will add it in the caption.*

Page 20, Line 6: "for the others, it was either the radon increase that was too low or the number of available hours" – Please clarify.

*We use different criteria to select the event, in particular, we check that there is an actual radon increase and we apply a threshold of 1 Bq m$-3$ and at least data spanning two hours (to have 4 datapoints minimum for the regression) as described in section 3.1.*

*We will reformulate for clarity.*

Page 21, Line 6:  "an underestimation for the higher ones" – Not clear.  RTM overestimates?

*Compared to one another, for the higher values, the inventory is lower than the RTM so underestimates the fluxes or depending on the point of view, the RTM overestimates them. We will reformulate for clarity.*

Page 21, Line 15:  "though soil chambers" – do you mean "through soil chambers"?

*We meant "using accumulation chambers", we will clarify the text.*

Page 21, Line 22:  "CO RTM and TNOf fluxes do not show a clear seasonal cycle or a trend over the period" – could you make a seasonality plot?Page 24, Line 1: "No trend is observed" – this is also mentioned elsewhere.  Did you try to do regression analysis?

*About these two points, we will apply the CCGCRV program to estimate trend and seasonality for all the species. We also plan to add the year 2023 and maybe 2024 if the radon flux map for that year gets available in the timeframe of our revision to improve the meaning of such trend estimation.*

Page 26, Line 3: do you mean "CH4, N2O and CO fluxes are in fair agreement with the inventories"?

*Yes, and we will reformulate accordingly.*

Code and data availability: the ICOS Carbon Portal address is not provided. Both the FLEXPART trajectories and the RTM code are not provided (shared on demand only) but should be archived in a public depository (e.g., https://zenodo.org/).

*We will add the address (https://meta.icos-cp.eu) and and make the code publicly available in a repository.*

**Minor comments:**

Page 3, Line 5: GHG and 222Rn "concentrations"?

Page 4, Line 27: "respectively, " – add "," before respectively (also check elsewhere in the text).

Section 2.2: Please add references for the Radon Tracer Method at the beginning of this section since this method has previously been used.

Page 8, Line 30: Please add "N" for latitude and "W" for longitude.

Page 9, Line 19: obtained BY

Page 10, Line 7: are showN.

Page 10, Line 15: remove the redundant "Bq".

Figure 18, Line 32: FEWER events

Figure 4 caption: using either….or both the maps and the footprints.

Page 24, Line 2-3: "we are looking here at nocturnal fluxes without photosynthesis only respiration" – how about "…without photosynthesis (i.e., with respiration only)"?

Page 24, Line 5, Line 14: "in average" should be "on average"; change "like" to "as".

*All the changes suggested above will be implemented.*

**Citation**:

Curcoll, R., Morguí, J.-A., Àgueda, A., Cañas, L., Borràs, S., Vargas, A., and Grossi, C.: Estimation of seasonal methane fluxes over a Mediterranean rice paddy area using the Radon Tracer Method (RTM), EGUsphere [preprint], https://doi.org/10.5194/egusphere-2024-1370, 2024.

Kikaj, D., Chung, E., Griffiths, A. D., Chambers, S. D., Forster, G., Wenger, A., Pickers, P., Rennick, C., O'Doherty, S., Pitt, J., Stanley, K., Young, D., Fleming, L. S., Adcock, K., Safi, E., and Arnold, T.: Direct high-precision radon quantification for interpreting high-frequency greenhouse gas measurements, Atmos. Meas. Tech., 18, 151–175, https://doi.org/10.5194/amt-18-151-2025, 2025.

Levin, I., Karstens, U., Hammer, S., DellaColetta, J., Maier, F., and Gachkivskyi, M.: Limitations of the radon tracer method (RTM) to estimate regional greenhouse gas (GHG) emissions – a case study for methane in Heidelberg, Atmospheric Chemistry and Physics, https://doi.org/10.5194/acp-21-17907-2021, 2021.

Thoning, K., Tans, P. P., and Komhyr, W. D.: Atmospheric carbon dioxide at Mauna Loa Observatory: 2. Analysis of the NOAA GMCC data, 1974–1985, Journal of Geophysical Research, https://doi.org/10.1029/jd094id06p08549, 1989.

Yver, C., Schmidt, M., Bousquet, P., Zahorowski, W., and Ramonet, M.: Estimation of the molecular hydrogen soil uptake and traffic emissions at a suburban site near Paris through hydrogen, carbon monoxide, and radon-222 semicontinuous measurements, Journal of Geophysical Research, https://doi.org/10.1029/2009jd012122, 2009.

---

## Author Comment (AC2)

**Six years of greenhouse gas fluxes at Saclay, France, estimated with the Radon Tracer Method**

Camille Yver-Kwok, Michel Ramonet, Léonard Rivier, Jinghui Lian, Claudia Grossi, Roger Curcoll, Dafina Kikaj, Edward Chung, and Ute Karstens

**Review 2**

https://doi.org/10.5194/egusphere-2024-3107-RC2

**General comments**

This manuscript reports on the use of the nocturnal-accumulation version of the radon tracer method (RTM), a tracer-ratio method for determining greenhouse gas emissions. In this study, radon-222 is used as a reference tracer, with known emissions, and the emission rate of several greenhouse gases is determined from the observed concentration ratio. Measurements are made over a six-year period from an inlet 100m above ground level.

The RTM has several unresolved issues. This is a fact that the authors recognise, citing Levin et al. (2021), but make the point that the benefits of the RTM make it worth exploring its application to the Saclay data set. The RTM is a relatively simple way to evaluate top-down greenhouse gas fluxes, without an inversion model. The method is therefore supported by a line of evidence which is independent from some of the uncertainties of transport models. I agree that this is a method worth exploring with these data and consider that the topic addressed by this manuscript is ultimately publication-worthy.

At this point in its development, though, there are three major issues which ought to be resolved before publication should be considered.

*We thank the reviewer for his helpful comments about the manuscript and for raising critical points to be improved. Our answers are shown below each point and in italic.*

**Specific comments**

[1] First, regarding methodology, I am concerned that the STILT footprints (shown in Fig 8) have been calculated or used inappropriately, although this might also be a misunderstanding on my part arising from an incomplete description of the methodology. The footprints are used to determine the influence region of nocturnal radon measurements and therefore to calculate a representative land-surface emission rate. The RTM concerns itself with the relative increase in radon concentration since the establishment of a stable nocturnal boundary layer in the late afternoon, at a time $t_0$. The nocturnal accumulation period lasts until the next morning, so therefore a measurement at time $t$ should have an associated radon flux which is calculated from

a footprint integrated over the period ($t_0$, $t$). In contrast, as far as I can discern, the STILT trajectories have been calculated from 10-day long retroplumes (i.e. backwards trajectories, generalised to account for dispersion), based on the information supplied by the Carbo Europe website, which is given as the source of these footprints. The use of a 10-d long retroplumes explains the very large influence region visible in Fig 8, much larger than even a 10 m/s flow would travel over ~ 10h (360 km, or roughly as far as the eastern border of France). In practice, because the RTM selects nights with relatively strong radon accumulation, RTM tends to bias towards calm nights and the main influence region should be smaller again. The footprint published by Levin et al. (2021), while not directly comparable as it is from a lower height above ground, covers a much smaller region, well constrained within a 150km x 150km box.

Staying on the topic of the footprint calculation, we also see that (1) Saclay is close to a local minimum in radon emissions, according to rightmost column of Fig. 8 yet (2) a comparison between radon emissions averaged over the STILT-calculated footprint vs the Saclay pixel, seen in Fig. 7, shows that the local radon emissions are (apparently) almost always higher than those averaged over the night footprint. I would expect the night footprint emissions to be distributed around the Saclay pixel, because there are higher radon emissions to the southwest but lower radon emissions from pixels immediately east of Saclay. My suspicion is that this seemingly contradictory result is from the use of footprints which extend too far back in time combined with the inclusion of ocean pixels in the calculation. One of the main assumptions of the RTM is that emissions of the tracer of interest are distributed over a similar geographic area as the emissions of radon, meaning that ocean fluxes (where radon emissions are vanishingly small) are out of scope for this method. As mentioned above, my understanding of the RTM is that the footprints should be recalculated with a much smaller integration time (meaning that oceanic pixels barely contribute to the calculation) but in addition, on the rare occasion when a backwards plume travels over the ocean, the radon flux should be calculated from a conditional average and only include land-surface points.

*As noted by the reviewer, we are indeed using the 10 days backward footprints. This was done as the idea was to propose a ready-to-use method and these are the footprints readily available on the ICOS Carbon portal. We assumed that the far-away contributions would be anyway almost null and so that the radon fluxes would be most influenced by the closer area around the station. To address properly this issue, we have now calculated for the year 2020, radon fluxes estimated with 5 hours, 10 hours, 24 hours, 48 hours, 96 hours and 240 hours retroplumes and could compared them. Especially in summer, it appeared that the difference between fluxes calculated with the 10 hours and the 240 hours retroplumes are significant.*

*Then, we, as suggested, calculated the average wind speed (6m/s) at Saclay during the nights used in the study over the whole 2017-2024 period. This value combined with the eight hours period that we study lead us to a 175 km radius around the station. We applied this mask to the 240 hours retroplumes and showed that the radon fluxes compare very well with the fluxes calculated with the 10hours retroplumes.*

*In the corrected manuscript, we will thus use this mask to reevaluate our radon fluxes and then the GHG fluxes which will a priori lead to higher fluxes especially in summer.*

[2] A second major concern is that the authors miss an opportunity to report on the observed trend in greenhouse gas emissions. One main finding of the present work, echoing what others have reported, is that the uncertainty in radon emissions is presently too high for the radon tracer method to be useful for absolute flux estimates. Consequently, it is important to report trends in emissions. Presently, the monthly mean fluxes are reported in Figs 11-13, but the data are noisy and the figure is insufficiently clear to draw conclusions. I recommend that the authors consider performing additional analysis to show a trend (naively, even extending the averaging period might be enough to better constrain the trend). If, with additional work, the RTM is not able to constrain a trend well enough to validate *a priori* trends in greenhouse gas fluxes then a conclusion stating this, while not a desirable outcome, would nevertheless be useful.

[3] My third concern, which is related to the previous one, is that the overall purpose of the manuscript is not altogether clear. At first glance, the mean GHG fluxes are the main result but the significance of these fluxes is undermined by the uncertainty in radon emissions, even for the most up-to-date radon emissions data (Karstens and Levin 2024). Other results from the manuscript include: a sensitivity analysis assessing the RTM, a brief comment on the VPRM biosphere exchange model, and a comparison between the reported fluxes previous studies. While these are all points worth discussing, I recommend that the main conclusion of the paper should be clarified. In my opinion, candidates for re-focusing the manuscript are either the trend in GHG emissions (observed vs. the inventories) or an analysis of the RTM, but ultimately this is a decision for the authors.

*Concerning the second and third points, we were reluctant to talk too much about trends as we have only 6 years of data (less for N2O). However, the focus of the paper, albeit obviously not clear enough was about the emissions and not about the limits of the methods that have been very well described before (Levin et al., 2021). To have a clearer focus, we will add 2023 (and 2024 if the radon flux map and footprints are available within the timeframe of the revision) for which we have GHG and radon data and exhalation map and add an analysis of the trends and seasonal cycles for each species.*

**Minor and technical comments**

Page 2 Line 22: "sophisticated atmospheric transport modelling" : it may be worth mentioning that the RTM provides a measurement which is independent of an atmospheric transport model, perhaps more important than the fact that it's easier to implement. (GHG fluxes are important enough to be worth measuring, even if the method is difficult)

*We will edit the text following your recommendation.*

P3 L3: My reading of the latest flux maps for Europe (Karstens and Levin 2024) is that the new maps still have ~factor of 2 uncertainty, whereas this text gives the impression of such a large uncertainty being a thing of the past.

*It is indeed true. However, for the Saclay site, a comparison between radon data and a model using these maps showed that the map using the GLDAS scheme was performing better, with a smaller bias and was therefore chosen for the study.*

P3 L13: The fact that radon emissions are variable in space and time is not especially new, e.g. Schery et al (1984). I also think the flux map from Zhou et al. (2008) deserves to be cited as an earlier example of a country-scale radon emissions map.

*It is true that this fact that the radon emissions are variable is not new though it was still assumed to be so in publications like Levin et al., 1999, Hammer and Levin, 2009, ... Flux maps began to be available around 2010ish. We will reformulate for clarity and add Zhou et al. in the probably non-exhaustive list of radon maps available.*

P3 L16: I would use the word "updated" instead of "improved" because it's not clear that the new maps have improved accuracy. The main benefit is that the updated map has a higher temporal resolution, as far as I understand it.

*We agree that the uncertainties of the radon flux maps are not substantially smaller than the previous version. Although the updated radon flux maps are based on newer and improved versions of soil moisture reanalysis, the relatively large difference between the two soil moisture reanalyses and the resulting difference in the radon fluxes implies that the appropriate term is "updated" rather than "improved". We will edit the text accordingly.*

P3 L25: "ICOS Class 1": Please explain the consequences of the tower being 'Class 1'

*Being an ICOS class 1 means measuring more parameters than in class 2, for example, CO is mandatory for class 1 while only CO and CH4 are mandatory for class 2. Class 1 is also measuring flasks. Details on the class' difference can be found in the ICOS Atmosphere Station specification (https://doi.org/10.18160/GK28-2188). The text will be edited for clarity.*

P4 L3: The Pal and Haeffelin study is unlikely to be a good source of information about Nocturnal Boundary Layers (NBLs) below 100m. Their study used a high-power research lidar, ALS-450, which (according to the manufacturer) has full optical overlap at ~300m. It is unlikely to be able to detect NBL below 100m, and indeed their figure shows that the measured NBL *never* drops below 100m, supporting the idea that the instrument may be unable to detect such low layers. Comparing radon concentration at two heights on the tower (or using other meteorological data from the tower itself) might be a useful alternative for quantifying how often the 100-m level decouples from surface emissions.

P4 L4: "most of the time", it would be better to quantify this, e.g as a percentage of nights, or mention that you quantify this (or alternatively, quantify the proportion of nights when the RTM was successfully used) later in the results

*Following your advice, we compared the nocturnal radon concentration at two heights, 15 m and 100 m from February 2022 to June 2025 (we have no data at 15 m before). We calculated their correlation. We found that 80% of the time, the two radon concentrations are well coupled and that the 100 m level stays below the boundary layer height. We will modify the text accordingly to use this metrics instead of the Pal and Haeffelin study.*

P4 L14: "The values found for SAC…" it is unclear whether this is a range which applies to both observations and models, or if one value is observations and the model. Also, I was confused as to why a model is involved if this is an observed flux? Please edit for clarity.

*The values are a range for both the measurements made at Saclay and the results from the flux map developed in Karstens et al., 2015 for the Saclay pixel. We will clarify the text.*

P4 L16: According to the website, the footprint function is a 10-day integral (i.e. particles are tracked 10-days backwards in time). For this study, the footprint for a nocturnal measurement should only be integrated backwards as far as the previous afternoon when the stable boundary layer began to establish itself.

*Please see answer to your main issue.*

P4 L28: Regarding the radon detector uncertainty, also quote the sensitivity (expected to be approx. 21 cpm/Bq/m3), based on Chambers et al. (2022) Table 1.

*We will do so.*

P5, Fig1: "…shows the absolute number of data…" By eye, the radial axis looks like windspeed for the GHG also, please double-check.

*The radial axis for all the plots are showing the number of data within the circular bins.*

P5, L3: "..well-mixed layer.." If it is a textbook case, the NBL will not be well mixed. The assumption of a well-mixed NBL is convenient for developing the RTM equation, but it is not, in fact, a necessary assumption. Here, one could develop the RTM by starting from the footprint-based analysis, as used in the FLEXPART and STILT models, or state here that the mathematical development is simplified – an 'illustrative' development of the method.

*Thanks for the suggestion, we will edit the text accordingly.*

P5, L5: Some terms in Eqn. (1) are not strictly defined ($\Delta t$ and $\Delta C$). I can guess that $\Delta t$ is the time since the establishment of a stable boundary layer, but equally it might just mean 'a small timestep'.

*We will clarify the text by adding the definition of the terms ($\Delta t$ being indeed the time since the establishment of a stable boundary layer and $\Delta C$ the temporal variation of the concentration over this period).*

P5, L5: The radon decay term is an approximation, unless $\Delta t \to 0$. It is reasonable to assume this, provided that $\Delta t \ll$ (the half-life of radon), but please note the approximation.

*We will do so on the revised text.*

P6 L13: "Hence, the radon flux…" Missing from this description is how far back in time the dispersion model is run for to calculate the S-R matrix, which should be only a few hours.

*We will clarify the text here to add the combination of the 10 days backward calculation with a mask to address this issue.*

P8 L18: "measurements from csv files": I don't understand the significance of this remark, please clarify

*This comment was added as the code was first developed to be 'plug-and-play' using only available data from the ICOS Carbon Portal. For the sensitivity part, however we wanted to be able to test different options. We will delete or reformulate for clarity.*

P8 L21: It is certainly important to take into account the radon detector's response time, however deconvolution is optional. The other option, since 1-minute GHG data are available, is to process the 1-minute data with a forward model of the radon detector's response. Computationally, and numerically, this is much simpler. Griffiths et al. (2016), Fig. 8, compares the two options. Either way, Griffiths et al. (2016) is an appropriate citation.

*We will add the citation in the revised version. As radon can be used by itself, e.g. to help diagnose models, we felt that it was better if the radon was corrected for its response time.*

P8 L22: Check that you're correcting to an appropriate reference. In a previous section, you convert GHG measurements from mixing ratio to concentration (presumably concentration at ambient temperature and pressure) so radon measurements should also be reported at ambient temperature and pressure.

*Thank you for the remark, this has been taken into account and will be clarified in the text.*

P8 L29: "The backtrajectories were calculated…". There are two time variables in the footprint calculation which should be distinguished (1) the measurement time, and (2) the initial time in the source-receptor relationship. Here, it is not clear which of these two times the "24-h window" is referring to.

P8 L25: FLEXPART-WRF is a version of FLEXPART which takes WRF-model meteorological fields as input, however it is stated that "This FLEXPART model uses ECWMF ERA5 meteorological files as inputs". This needs clarification.

*The paragraph will be changed to make it more clear:*

*"The FLEXPART-WRF model version 3.3.2 (Brioude et al., 2013) run at the Universitat Politècnica de Catalunya (UPC, Spain), is used here. This FLEXPART model uses WRF v4.2.1 output files as inputs for its back trajectory calculations. ECWMF ERA5 meteorological files were used as initial and lateral boundary conditions for WRF. This model was used with an output resolution of 0.05 degrees in order to fit with the new ERA5-Land and GLDAS-Noah2.1 radon maps. The back trajectories were calculated for a 24h window time, beginning at 00:00 UTC and assuming the 0-100 m layer as the footprint layer. For the Saclay site, the spatial domain used was 42.9°–54.5° in latitude and -6°–16.2° in longitude."*

P8 L29: "0-100m layer" This choice is not unreasonable, but it does make it impossible for the model to indicate when the modelled atmosphere at 100m AGL is decoupled from the surface.

*This is indeed a limitation. However, as commented above (P4,L3-4), we showed that 80% of the nights at Saclay, the 100m level is coupled with the lower level and thus the model assumption should be valid for most of the two months studied with this model.*

P9 L15: Units of Bq/m2/h are used here for radon flux, but mBq/m2/s elsewhere. Please standardise to one or the other, and also note the justification for this value at this point in the text. It would also be helpful to indicate how this value compares to the European mean.

*We could not find units in mBq/m2/s in our manuscript but noticed that in some of the figures h and m2 were inverted. We will standardise them. We will also clarify why we use 52 Bq/m2/h which comes from Yver et al., 2009.*

P10 L1: Here an R2 value of 0.6 is mentioned, which seems overly permissive and perhaps arbitrary? Is it possible to show how the flux estimate changes as a function of the R2 cut-off?

*The 0.6 value was chosen as a compromise between a high number of events and a very high correlation. It was used previously in Hammer et al;, 2009 and Yver et al., 2009. We will add a plot showing the average flux in function of the R2 cut-off.*

P11 Fig 3: "From the literature": use a citation (abbreviated if necessary)

*We will add the citation (Yver et al., 2009)*

P11 Fig 3: "User Rn flux": replace with a more meaningful label

*We will replace by 'Flux from Yver et al., 2009*

P11 L7: "The standardization…" Consider not showing the non-standardised case. There is no question that the difference in instrument response should be taken into account before calculating a correlation. A more meaningful comparison would be between (1) deconvolution applied to the radon measurements or (2) the radon detector response function applied to the CO2/CH4 measurements. Or move this to a supplement.

*We chose to add this discussion and result as new users could not be well aware of the time response issue or think that it is not really significant and then produce biases numbers.*

*We will move it to the supplement.*

P11 L12: "The transport models…" Edit this sentence for clarity

*We will change the sentence to "The transport models used here have less impact. Moreover, using a simplified decay term compared to have this term included in the model leads to insignificant differences."*

P12 L9: How are ocean pixels handled? If only land-based sources/sinks are being considered (and oceanic air is assumed to be 'background' concentrations) then the calculation should be a conditional average. That is, select land points, and take a weighted average of radon flux (weighted by the footprint).

*In the ocean pixels, the radon fluxes are null so their weight in the footprint does not matter. However, in the revised version, using a mask, we will have no more ocean pixel.*

P13 Fig. 5. There are outlier points, one negative and one high, in this plot. These might be worth remarking on in the text; is there anything unusual about the outlier point, around the 21st

in winter?  Or the negative flux, seen in summer?  Do either of these cases point to the assumptions of the RTM not being upheld?

*We will comment on the events in the text. In winter, on the 02/21, we do have high concentration of methane at the beginning of the night but with a R2 of 0.76 and high concentration of CO2 and CO as well, it looks like a synoptic event with a polluted airmass. In summer on August 3rd, we observe low winds, a relatively strong increase of the radon while the other greenhouse gases stay relatively stable, the R2 is at 0.66.*

P15 Fig. 7: I assume that this the same map as the one referred to as "GLDAS Noah" in Fig 8, please use standard nomenclature.

*We will correct the caption and the legend in the figure accordingly.*

P15 Fig. 7: As mentioned in the Major Comments, from looking at the STILT footprint map, it looks like there should be periods (especially in Summer) when the airmass comes from the SW and passes over radon emissions of > 0.06 Bq/m2/sec (216 Bq/m2/h)

*Following the answer to the major comment, this figure will be redone with the masked footprints.*

P16 Fig. 8: Include map scale (e.g. with scale bar or by labelling lat/lon) and projection; Units are inconsistent with usage elsewhere; a linear colour scale would be more appropriate for this usage of the footprint OR include a contour which encloses 99% (or some other large fraction) of the footprint.

*We will edit the map according to your recommendation and add the square that defines our nocturnal footprint.*

P17 L8: "However the air measured…" This would be straightforward to resolve by configuring STILT to create a footprint which has hourly-resolved emission time (that is, a 4 dimensional array with dimensions of (measurement time, emission time, latitude, longitude) ).  Recording the footprint in this way would also allow the straightforward calculation of radon decay during post-processing.

*As said previously, we chose to use tools readily available and unfortunately, we do not have access to hourly-resolved footprint by default on the CP. We will edit the text to note that, with a time-resolved footprint, this would be easier.*

P18 L17: "total uncertainty" is mentioned, please define how uncertainty is parameterised (e.g. one standard deviation, 95% CI, etc.)

*To calculate the total uncertainty, we propagate the errors with a standard summation in quadrature. The errors on the slopes and on the greenhouse gas measurements are standard deviation of the estimated parameters.The uncertainty on the model comes from Karstens et al., 2015, and is itself calculated as the propagation of errors from different sources. They stem mostly from the differences between the model and observed values. The error for our detector os described in more details in Grossi et al., 2020, as the combination of a calibration source uncertainty of 4 %, a coefficient of variability of valid monthly calibrations of 2 %–6 %, and a counting uncertainty of around 2 % for radon concentrations $\geq 1$ Bq m$^{-3}$.*

P18 L28: Is it valid to present averages?  That is, do you believe that outliers are caused by emission events, or would it be more appropriate to present a trimmed mean?  The data, after all, are rather skewed.

*Depending on the species, this may indeed not always be the best metrics. N2O outliers are most probably caused by short-lived emission events and indeed skew the average. This why we show the variability in the figures and discuss it in the text. The averages allow however a quick comparison with the inventories.*

P19 Fig 10: Not clear (from the figure caption) what time period each of the dots represent (probably nightly?)

*Each of the dots represent a nocturnal flux. This will be clarified.*

P20 L1: "Boiler Room", if possible be specific and name the facility. Is this a large-scale "district heating" facility, perhaps?

*This boiler room is the heating facility for our research center so small-scale but very local. We will edit the text for clarity.*

P20 L7: "…the correlation was too low…" It might be helpful to discuss this further, as it's a limitation of the RTM.  Was this a period of strong winds, for instance?  Or a period of calm conditions and stagnation?

*We will extend the discussion in the revised version. For most of the month, we have relatively strong winds between 5 and 15 m/s which lead to an accumulation of radon below our threshold even if for some the correlation was good enough.*

P20 L15: Is the purpose to choose a winner out of EDGAR and TNO?  This section seems to \*almost\* make this point, but could be more clear.  It is worth stating, even if the conclusion is the unsatisfactory one that the RTM is too ambiguous to choose.

*The purpose was to discuss the inventories not to pick a winner which for only one station would not be very significant. We will edit the text to reflect however which inventory seems to agree better four our particular site.*

P21 L27: Is the 'boiler room', discussed as part of the CH4 section, a potential source of CO?

*No peaks of CO were observed during the boiler room CH4 peaks.*

P24 L3: "…nocturnal fluxes without photosynthesis …" I agree with this statement, but it seems to contradict an earlier statement that "…, the air measured at midnight at Saclay travelled during the day and it is thus not straightforward to select which hours of the inventory should be favored."

*We will reformulate the first part of the text in the revised version of the manuscript. By applying the mask, we are constraining most of the data to a nocturnal footprint, indeed without photosynthesis especially in winter. However, for the TNO fluxes, as we will still be working with aggregated footprints, applying a daily cycle stays complicated.*

**Technical comments**

P21 L28: "cadran" -> "quadrant"

P24 L18: "1.5" -> "1.5 x 10^-3"

P6 L5: "in concentrations" should be "into concentrations".  As an aside, I believe that the opposite conversion would also work (converting radon into a mixing ratio).

P10 L11: consider "soil moisture" rather than "humidity", as the latter sounds related to water vapour

Some acronyms need explaining, e.g. CCGCRV, NRT

*All these modifications will be applied.*

The impact of this publication would be increased if the datasets and code used to generate the results in the paper were made available more conveniently, for instance by depositing them with a repository like zenodo.

*We will make the code publicly available in a repository.*

**References (cited in this review)**

Brioude, J., Arnold, D., Stohl, A., Cassiani, M., Morton, D., Seibert, P., Angevine, W. M., Evan, S., Dingwell, A., Fast, J. D., Easter, R. C., Pisso, I., Burkhart, J. F., and Wotawa, G.: The Lagrangian particle dispersion model FLEXPART-WRF version 3.1, Geoscientific Model Development, https://doi.org/10.5194/gmd-6-1889-2013, 2013.

Chambers, S. D., Griffiths, A. D., Williams, A. G., Sisoutham, O., Morosh, V., Röttger, S., Mertes, F., and Röttger, A.: Portable two-filter dual-flow-loop $^{222}$Rn detector: stand-alone monitor and calibration transfer device, Adv. Geosci., 57, 63–80, https://doi.org/10.5194/adgeo-57-63-2022, 2022

Griffiths, A. D., Chambers, S. D., Williams, A. G., and Werczynski, S.: Increasing the accuracy and temporal resolution of two-filter radon–222 measurements by correcting for the instrument response, Atmos. Meas. Tech., 9, 2689–2707, https://doi.org/10.5194/amt-9-2689-2016, 2016

Grossi, C., Chambers, S. D., Llido, O., Vogel, F. R., Kazan, V., Capuana, A., Werczynski, S., Curcoll, R., Delmotte, M., Vargas, A., Morguí, J.-A., Levin, I., and Ramonet, M.: Intercomparison study of atmospheric $^{222}$Rn and $^{222}$Rn progeny monitors, Atmos. Meas. Tech., 13, 2241–2255, https://doi.org/10.5194/amt-13-2241-2020, 2020.

Hammer, S. and Levin, I.: Seasonal variation of the molecular hydrogen uptake by soils inferred from continuous atmospheric observations in Heidelberg, southwest Germany., Tellus B, https://doi.org/10.1111/j.1600-0889.2009.00417.x, 2009.

Karstens, U., Schwingshackl, C., Schmithüsen, D., and Levin, I.: A process-based 222radon flux map for Europe and its comparison to long-term observations, Atmospheric Chemistry and Physics, https://doi.org/10.5194/acp-15-12845-2015, 2015

Levin, I., Karstens, U., Hammer, S., DellaColetta, J., Maier, F., and Gachkivskyi, M.: Limitations of the radon tracer method (RTM) to estimate regional greenhouse gas (GHG) emissions – a case study for

methane in Heidelberg, Atmospheric Chemistry and Physics, https://doi.org/10.5194/acp-21-17907-2021, 2021.

Pal, S. and Haeffelin, M.: Forcing mechanisms governing diurnal, seasonal, and interannual variability in the boundary layer depths: Five years of continuous lidar observations over a suburban site near Paris, Journal of Geophysical Research, https://doi.org/10.1002/2015jd023268, 2015

Schery, S. D., D. H. Gaeddert, M. H. Wilkening, Factors affecting exhalation of radon from a gravelly sandy loam, J. Geophys. Res., 89, 7299–7309, 1984.

Yver, C., Schmidt, M., Bousquet, P., Zahorowski, W., and Ramonet, M.: Estimation of the molecular hydrogen soil uptake and traffic emissions at a suburban site near Paris through hydrogen, carbon monoxide, and radon-222 semicontinuous measurements, Journal of Geophysical Research, https://doi.org/10.1029/2009jd012122, 2009.

Zhuo, W., Guo, Q., Chen, B., and Cheng, G.: Estimating the amount and distribution of radon flux density from the soil surface in China, J. Environ. Radioact., 99, 1143–1148, doi: 10.1016/j.jenvrad.2008.01.011, 2008.

---

## Author Comment (AC3)

**Answers to Editor Comments**

*We thank the editor for her remarks that are helpful to improve the manuscript.*

Line numbers refer to version 3 of the manuscript.

-- The first reviewer asked for literature values in he abstract. While I think the abstract is not the correct part of the text for further details, this should be incorporated in the introduction. However, I could not find a corresponding modification of the text.

*We also felt that there was no place for the reference in the abstract and let them in the general text when comparing with the result. We can add them in the introduction.*

*Page 2 l 31 "The RTM has been used in many studies to evaluate the fluxes between the atmosphere and ecosystems of trace gases such as $CO_2$, $CH_4$, $N_2O$, $H_2$ or COS (e.g.: Levin et al. (1999); Schmidt et al. (2001); Biraud et al. (2000); Messager (2007); Yver et al. (2009); Hammer and Levin (2009); Lopez et al. (2012); Vogel et al. (2012); Belviso et al. (2013); Grossi et al. (2018); Belviso et al. (2020); Levin et al. (2021); Tong et al. (2023)).*

*For example, Levin et al. (2021) calculated an estimate of 0.8 mg $CH_4$ $m^{-2}h^{-1}$ for the Heidelberg region between 2015 and 2020 while in Cabauw between 2016 and 2018 (Tong et al., 2023), the estimate reached 1.4 mg $CH_4$ $m^{-2}h^{-1}$ and 0.046 mg $N_2O$ $m^{-2}h^{-1}$ for $CH_4$ and $N_2O$ respectively. At Gif-sur-Yvette, neighbouring Saclay, values of 0.8 mg $CH_4$ $m^{-2}h^{-1}$, 545 mg $CO_2$ $m^{-2}h^{-1}$ and 0.068 mg $N_2O$ $m^{-2}h^{-1}$ for $CH_4$, $CO_2$ and $N_2O$ respectively, were found for the period 2002-2007 (Messager, 2007). In Lopez et al. (2012), $N_2O$ values were found for the same site within the range of 0.039 to 0.058 mg $N_2O$ $m^{-2}h^{-1}$ over the period 2002-2011. For CO, Messager (2007) found an average value of 1.46 mg CO $m^{-2}h^{-1}$ for Europe using measurement from Mace Head, Ireland over the period 1996-2006 with a tendency to decrease over time."*

-- The link to the ICOS data given in the data availability statement did not work when I tried.

*Indeed, the link seems to have slightly changed, we will correct it with the home page link: https://www.icos-cp.eu/*

-- The website for the NOAA curvefit code should be properly cited

*We will add the link to the documentation in the Figure 1 caption (https://gml.noaa.gov/ccgg/mbl/crvfit/crvfit.html)*

-- The previous Figure 6 seems to have been removed in the revised version but I could not find an explanation in the responses for that. Ther's a statement in the response to reviewer #1 that states a modifictaion of the caption.

*Following the second reviewer advice, we decided to show the results using only the properly treated radon data (i.e. with the correction for the time response). This made the Figure 6*

*superfluous. Contrary to our answer, we however decided against having supplementary materials when revising the manuscript.*

-- P12, Lines 29/30 the revised wording "that lead the days to fail to pass the criteria" does not make sense.

*We propose the reformulation:*

*"For this month, the correlation was too low for 19 days and for the other non-selected days, it was either the radon increase or the number of available hours that were too low (below 1  Bq m−3 for the radon increase or less than 2 hours of duration for the available data). "*

-- P13/L23: "an underestimation of the inventory for the higher ones" - the revised wording is as unckear as the previous one that the reviewer asked about.

*We propose the reformulation:*

*"The top right panel shows indeed a better correlation between RTM and inventory values for the lower RTM values (< 0.1 mg N20 m−2h−1 ) but systematic lower values of the inventory compared to the RTM values for the higher RTM values (>0.1 mg N20 m−2h−1 )."*

-- Reviewer 1 has suggested to provide a seasonality plot, but your respnse does not comment on this suggestion. Could you please provide the plot in your response, even if it does not go into the manuscript?

*Figures 11 to 14 were modified so that the left lower panel is showing the seasonality plots.*

-- Reviewer 2 suggested Zhou et al 2008 as an additional refernce which is now cited as Zhuo et al 2008. Please double check which spelling is correct.

*I got the citation directly from the article publisher and double-checked. It is Zhuo et al.*

-- P3/L31: Reviewer 2 asked for an explanation of the ICOS site classes. Why do you state that "only" CO and CH4 are mandatory for class 2 sites? The previous sentence states that class 2 sites have a larger number of compounds required than class 1 sites.

*We propose the reformulation:*

 *"For example, CO2, CH4  and CO are mandatory for class 1 while only CO and CH4 are mandatory for class 2. "*

--P4/L20: Reviewer 2 stated that the previous wording was not clear regarding where the given range applies to both the observations and further data. The new wording "for the observations made there and for the exhalation maps that used these measurements for verification. " still does not clarify this. Please state the range separately for the observations and the derived maps.

*We propose the reformulation*

 *"The values found for SAC for the direct measurements were 18 − 54 Bq m−2h−1. The simulations yielded values in the same range."*

-- Reviewer 2 asked about the mixing ratio/ conentration conversion (referring to P8/L22 ov the first manuscript version). Your response states a corresponding change to the text which I was unable to identify. Please clarify.

*We answered that point p6 l6-7: "We use the molar volume at 288.15 K and 101 325 Pa to match the radon data treatment defined in Kikaj et al. (2025)."*

-- Reviewer 2 enquired about value of 0.6 for r² and how the flux estimates change depending on the cjosen cut-off value of r². In your response you refer to a figure added to the manuscript which I could not find in the revised version. Please clarify which changes were mane to the manuscript.

*We added Figure 7 in the revised version that shows the radon flux in funtion of the cut-off value.*

*The figure is commented p10 l6-9: "To check the effect of the chosen R2 cut-off value, we have calculated the average radon flux for each run depending on which R2 cut-off we chose, from 0.4 to 0.8. This is shown on Figure 7. Depending on the R2 cut-off value, the average flux is varying, however, considering the standard deviation, the variations are not significant and confirm that a cut-off value of 0.6 is a good compromise between high correlation and number of events selected."*

-- Your response to reviewer 2 states moving content to supplements, however, there is not supplementary information submitted with the revised manuscript. Please clarify.

*When we submitted our answers to the reviewers, we were still reprocessing data and not modifying the manuscript alongside the answers. After careful considerations of the reviewer comments, we agreed that the radon "raw" data were not necessary and chose not to include them even in supplemental material.*

-- Your response states that code will be made publicly available, however, there is no mentioning of this in the data availability setion of the revised manuscript. Please add.

*We are working with the Carbon portal to make the code available. We plan to have the code reachable and usable on the Carbon Jupyter Notebook so it involves several steps. We should be able to provide a proper link before final publication.*

-- For the new version of what is now Figure 9, I was wondering if it still makes sense to keep the zoomed out/zoomed in panels in the two left columns. I think one columns with maps of an intermediate zoom level would be sufficient and would simplify the figure.

*We can remove the left column to simplify the figure.*

-- Figure 10 of the revised manuscript does not have a figure caption.

*The caption that is typed in the laTEX file should read:"Soil moisture content for the pixel containing SAC as modeled in GLDAS-Noahv2.1 from 2017 to 2025 with a porosity (noted por on the figure) of 0.439." We will investigate why it does not show in the compiled file.*

**Citation**: https://doi.org/10.5194/egusphere-2024-3107-EC1